# Effects of stochastic coding on olfactory discrimination in flies and mice

**Shyam Srinivasan**[1,2]*, **Simon Daste**[3,4], **Mehrab N. Modi**[5], **Glenn C. Turner**[5], **Alexander Fleischmann**[3,4], **Saket Navlakha**[6]*

**1** Kavli Institute for Brain and Mind, University of California, San Diego, California, United States of America, **2** Salk Institute for Biological Studies, La Jolla, California, United States of America, **3** Department of Neuroscience, Division of Biology and Medicine, Brown University, Providence, Rhode Island, United States of America, **4** Robert J. and Nancy D. Carney Institute for Brain Science, Brown University, Providence, Rhode Island, United States of America, **5** Janelia Research Campus, Howard Hughes Medical Institute, Ashburn, Virginia, United States of America, **6** Simons Center for Quantitative Biology, Cold Spring Harbor Laboratory, Cold Spring Harbor, New York, United States of America

* shyams@uci.edu (SS); navlakha@cshl.edu (SN)

**Data Availability Statement:** The mouse and fly datasets used in this paper are available at the following sites: Mouse: https://doi.org/10.48324/dandi.000167/0.220928.1306 The repository contains the main dataset, which is the one used in

## Abstract

Sparse coding can improve discrimination of sensory stimuli by reducing overlap between their representations. Two factors, however, can offset sparse coding's benefits: similar sensory stimuli have significant overlap and responses vary across trials. To elucidate the effects of these 2 factors, we analyzed odor responses in the fly and mouse olfactory regions implicated in learning and discrimination—the mushroom body (MB) and the piriform cortex (PCx). We found that neuronal responses fall along a continuum from extremely reliable across trials to extremely variable or stochastic. Computationally, we show that the observed variability arises from noise within central circuits rather than sensory noise. We propose this coding scheme to be advantageous for coarse- and fine-odor discrimination. More reliable cells enable quick discrimination between dissimilar odors. For similar odors, however, these cells overlap and do not provide distinguishing information. By contrast, more unreliable cells are decorrelated for similar odors, providing distinguishing information, though these benefits only accrue with extended training with more trials. Overall, we have uncovered a conserved, stochastic coding scheme in vertebrates and invertebrates, and we identify a candidate mechanism, based on variability in a winner-take-all (WTA) inhibitory circuit, that improves discrimination with training.

## Introduction

There are 2 often-cited features of neural representations. The first is that only a small set of neurons fire in response to each stimulus (called sparse coding), which aids discrimination. The second is that a somewhat different set of neurons respond each time the same stimulus is presented (called trial-to-trial variability), which could hinder discrimination. Here, we seek to understand how these 2 features can work together to enable both coarse and fine odor discrimination.

all the main figures, and a supplementary set comprising 4 datasets, which were used in reporting statistics in Table E in S1 Text. The main dataset is 164, and the supplementary datasets are 163, 7, 8, and 9. The statistics for all datasets are presented in Table E in S1 Text. Fly: https://doi.org/10.5281/zenodo.8166598 The repository contains the main dataset, which is the one used in all the main figures, and a supplementary set comprising 7 datasets, which were also used in Figs 4, 6, Fig C in S1 Text and Fig F in S1 Text as well as Table D in S1 Text. The main dataset is labelled accordingly, and the supplementary datasets are 110107_1, 110108, 110109_1, 110109_2, 09042009, and 110106_4. Besides the main dataset, Fig 4 used datasets 110107_1, 110108, 110109_1, 110109_2, 09042009, Fig 6 used datasets 110107_1, 110108, 110109_1, 110109_2, 09042009, and Fig F in S1 Text used dataset 110106_4. The statistics for all datasets (except 110106_4) are presented in Table D in S1 Text. The repository describes the layout and details of the datasets. Mouse time lapse video of PCx cell responses: https://dandiarchive.org/dandiset/000167/draft/files?location=sub-164%2Fsub-164_ses-20200124T161514_image%2Bophys Software for generating the model, and for analysis: https://doi.org/10.6084/m9.figshare.23994708 The code for generating the model and performing analysis on the data is present in figshare, which also contains a link to the github source code.

**Funding:** Pew Charitable Trusts - SN National Institutes of Health 1R01DC017695 and 1UF1NS111692-01 - SN Simons Center for Quantitative Biology at Cold Spring Harbor Laboratory - SN National Institutes of Health 1R01DC017437-03, 1U19NS112953, and S10OD025181 - AF CRCNS – ANR-17-NEUC-0002-01 - AF Robert J and Nancy D Carney Institute for Brain Science - AF Kavli Institute for Brain and Mind - SS Howard Hughes Medical Institute - GT and MNM The funders did not play a role in the design, data collection and analysis, decision to publish, or preparation of the manuscript.

**Competing interests:** The authors have declared that no competing interests exist.

**Abbreviations:** APL, anterior paired lateral; AUC, area under the curve; CS, conditioned stimulus; CV, coefficient of variation; DAN, dopaminergic neuron; FF, Fano factor; FOV, field of view; IACUC, Institutional Animal Care and Use Committee; KC, Kenyon cell; kNN, k nearest neighbor; LDA, linear discriminant analysis; MB, mushroom body; MBON, MB output neuron; OB, olfactory bulb;

Sparse coding has emerged as an important principle of neural computation [1,2]. Sparse coding can improve discrimination by reducing overlap between representations [3] and leads to more efficient computation because fewer active neurons implies less energy expenditure [4]. Sparse coding has been observed across sensory modalities, from vision and audition to olfaction [5], and across species, from vertebrates [6,7] to invertebrates [3,8–11].

A closer examination of stimulus responses, however, poses a challenge for our view of sparse coding's role in the brain. Most studies report averaged or thresholded stimulus responses across trials, which result in an underestimate of the responsive population. For example, in the olfactory system of fruit flies, the percentage of higher order neurons that respond to an odor can be as high as 15% per trial [9,12], as opposed to 5% if averaged and thresholded across trials. Similarly, in the mammalian olfactory system, at least 16% of the coding population responds per trial in rats [13] and about 20% responds in mice [14], up from 10% [6] when averaged across trials. These studies show a discrepancy in perceived sparsity as calculated from multiple trials versus single trials. Explanations of animal behavior need to account for this discrepancy [15].

To explain cognitive processes downstream of peripheral sensory coding, neuroscientists must contend with a second issue that is closely tied to the problem of overestimating sparsity: variability in neural responses across trials. Trial-to-trial variability is ubiquitous in the brain. Among sensory regions, it is observed in the visual system [16–18], the auditory system [19], the whisker thalamus [20], and the motor system [21,22]. Within learning and memory systems, it is observed in the hippocampus and the entorhinal cortex [23], the prefrontal cortex [24], and the basal ganglia [25,26]. Traditionally, variability is viewed as a byproduct of sensory-level noise and is thus a "nuisance" the brain must contend with. Analyses that threshold or average responses across trials effectively support this view. In contrast, variability can be beneficial in some situations [27,28], suggesting that variability may be intrinsic to the circuit. Variability and lower sparsity begs the question of how animals can produce robust decisions from a single exposure of a sensory stimulus. One can envision real-world scenarios, e.g., sensing a predator, that might not afford the luxury of multiple trials for decision-making.

To understand how sparsity levels and variability affect neural ensemble coding and decision-making, 4 questions need to be resolved. First, what percentage of cells respond consistently and inconsistently across trials? Second, are there differences in the information encoded by these 2 populations of cells? Third, what are the mechanisms that produce trial-to-trial variability? Fourth, do cells with different levels of variability make different contributions to odor discrimination?

Our goal in this study is to characterize sparsity and variability of odor responses in the fly mushroom body (MB) and the mouse piriform cortex (PCx), and to understand the consequences of these factors towards discrimination and learning. The olfactory system is attractive to investigate these questions because it exhibits both sparsity and variability. In addition, as MB and PCx are only 2 synapses from the external environment, circuit anatomy and physiology linking sensory responses to neural coding and behavior are well defined, particularly in flies [29–31] and conserved in mammals [32–34]. We show, in both flies (using existing MB data [9]) and mice (using new PCx data collected for this study), that cells responding to an odor fall along a continuum between 2 extremes: from a small number of highly reliable cells that respond in every trial, to a large number of very unreliable cells that respond only in one out of many trials. We show that reliable cells can better decode odor identity than unreliable cells, suggesting that reliable cells represent a more stable (i.e., a less stochastic) component of the odor code. However, we show that highly reliable cells by themselves largely overlap for similar odors, whereas highly unreliable cells do not. Thus, while highly reliable cells can easily distinguish dissimilar odors, we propose that more unreliable cells can, with training, be used

OSN, olfactory sensory neuron; PCx, piriform cortex; PN, projection neuron; ppm, parts per million; ROI, region of interest; SEM, standard error of the mean; SR, stochastic resonance; SVM, support vector machines; US, unconditioned stimulus; WTA, winner-take-all.

to distinguish very similar odors. Finally, using a computational model of the olfactory circuit, we show that the variability observed experimentally far exceeds what would be expected under reasonable levels of sensory noise and that variability (noise) in the winner-take-all (WTA) inhibitory circuit is a more likely mechanism for generating stochastic codes.

## Results

### Basic anatomy and physiology of the olfactory circuit

In flies and mice, the olfactory circuit comprises 3 stages (Fig 1A), with key elements of the circuit architecture being conserved across species [35]. In stage 1, odor information is captured by olfactory sensory neurons (OSNs) in the mammalian nose [36–38] or fly antenna and maxillary palp [39,40], where each OSN type expresses a single receptor type that preferentially responds to a particular chemical class. There are about 1,000 OSN types in mice and 50 types in flies, which are responsible for sensing the vast space of possible chemical compounds. Thus, odors are encoded as a combinatorial code; i.e., every odor is represented by a unique combination of neurons that respond to that odor [41–43]. In stage 2, odor information captured by OSNs is transferred to glomeruli in the olfactory bulb (OB) of mammals [44–46] or antennal lobe of flies [40]. Glomerular encoding is modified by a lateral inhibition circuit, which increases the reliability of the odor representation and normalizes its output range [47,48]. In stage 3, glomeruli, through projection neurons (PNs), pass odor information to PCx cells in mice or MB Kenyon cells (KCs) in flies [31,32]. In flies, a WTA circuit sparsens KC odor responses (Fig 1A); all KCs activate a giant inhibitory neuron (called the anterior paired lateral (APL) neuron), which then negatively feeds back to suppress the activity of less responsive KCs [3]. PCx also contains a WTA circuit, wherein PCx principal cells activate inhibitory neurons in layers 2 and 3 of PCx, which then negatively feedback to suppress less responsive PCx cells [32,49,50]. The sparse encoding of the odor in the third stage (MB or PCx) is used downstream by the animal for olfactory discrimination and learning [31,51,52].

Here, we examined the sparsity and variability of responses to odors in the third stage of the circuit.

### Reliable and unreliable components of odor representations in fly mushroom body and mouse piriform cortex

**Fly.** The original studies of population activity in KCs measured sparseness using a particular response criteria [9]. They showed that while odor responses in the MB evoked consistent activity in about 5% of KCs across trials, the number of cells active per trial was much higher at about 12% (Methods, Data Analysis). Here, we used those same criteria to group cells into reliable and unreliable cells, as illustrated in Fig 1B. Reliable cells are those that respond in more than half the trials (e.g., cell 1 with a reliability of 3 out of 4 trials), and unreliable cells are those that respond in ≤ half of the trials (e.g., cell 3 with a reliability of 1 out of 4 trials). The reliability property of a cell is defined to be the number of responsive trials. We applied these criteria to the dataset—containing responses of 124 KCs in 1 fly to 7 odors with 6 trials per odor—to investigate the sparsity and variability of odor responses in the MB. This experiment was repeated in 5 other flies with similar results (Methods, Table D in S1 Text). This division of cells into 2 classes (reliable and unreliable) is used to ease exposition and to serve as a comparison to earlier work [8,9]; in the next section, we show that these cells actually fall along a reliability continuum.

Fig 1B illustrates how the responses of KCs vary across trials. On an average trial, about 12% of the 124 KCs exhibited a significant response, with 5% of those cells being reliable and

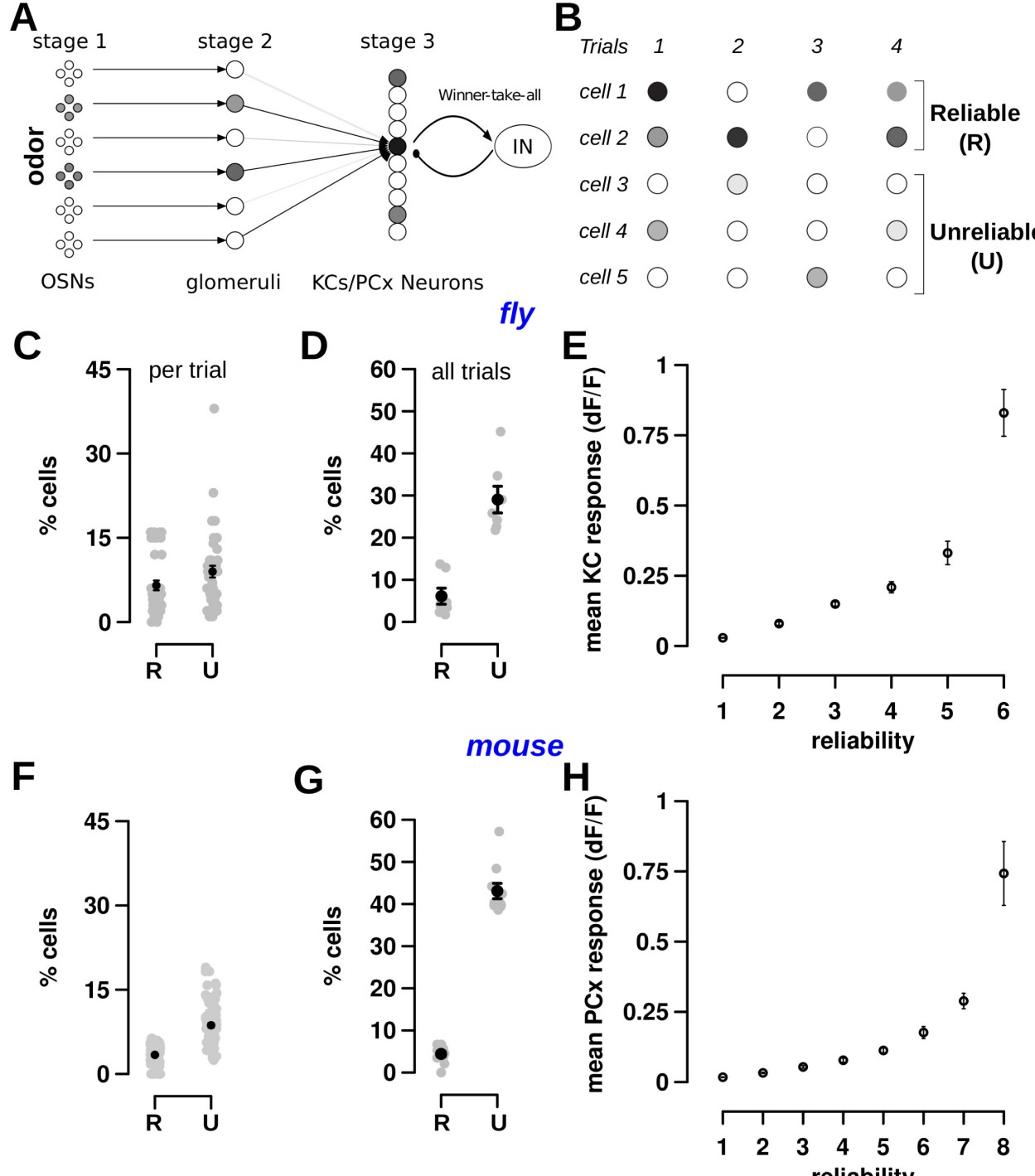

**Fig 1. Response reliability is similar in third-order neurons of the fly and mouse olfactory circuit.** (A) Schematic of the first 3 stages of fly and mouse olfactory circuits. In stage 1, odors are sensed by OSNs. In stage 2, OSNs pass odor information onto glomeruli structures in the antenna lobe (flies) or olfactory bulb (mice). In stage 3, odor information is passed to a larger number of KCs in the MB (flies) or cells in the PCx (mouse). Stage 3 cells are sparsified by an inhibitory WTA circuit, which suppresses the activity of less responsive cells. (B) An example schematic showing response variability in stage 3 cells. Some cells (cells 1 and 2) have a large response in most trials (indicated by R for reliable cells), while most cells respond in ≤ half the trials with a smaller response (indicated by U for unreliable cells). (C–E) Responses of 124 KCs in flies for 7 odors with 6 trials per odor; data from [9]. (C) The percentage of responsive KCs that are reliable (5.26 ± 0.7%) and unreliable (7.23 ± 0.84%) per odor trial. There are 42 gray points, 1 per odor-trial pair. (D) The percentage of KCs that are reliable (6.1 ± 1.9%) and unreliable (29.0 ± 3.16%) across all trials. Each gray point is an odor and the black point represents the mean. There are many more unreliable cells than reliable cells. (E) The mean response of a KC (y-axis) increases with the cell's reliability (x-axis). The measure of the fit, using linear regression on a log-plot, is shown in panel A in Fig A in S1

Text ($r = 0.97$). (F–H) Responses of 285 PCx cells in mice for 10 odors with 8 trials per odor; data collected in this study. (F) The percentage of responsive PCx cells that are reliable ($3.3 \pm 0.2\%$) and unreliable ($8.7 \pm 0.4\%$) per trial. There are 80 gray points, 1 per odor-trial pair. (G) The percentage of PCx cells that are reliable ($4.4 \pm 0.7\%$) and unreliable ($43.1 \pm 1.8\%$) across all trials. Again, there are many more unreliable cells than reliable cells. (H) The mean response of a PCx cell (y-axis) increases with the cell's reliability (x-axis). The measure of the fit is shown in panel B in Fig A in S1 Text ($r = 0.98$). Error bars in all panels indicate mean ± SEM. Datasets used in these plots for flies (A–E) and mice (F–H) are stored within the Zenodo and Dandi repositories as the fly main dataset and dataset 164, respectively. Links on accessing the data are in the Data Availability section, and the data repositories contain further details on the datasets. The data underlying the graphs shown in the figure can be found in S1 Data. KC, Kenyon cell; MB, mushroom body; OSN, olfactory sensory neuron; PCx, piriform cortex; SEM, standard error of the mean; WTA, winner-take-all.

7% being unreliable (Fig 1C). By definition, the subset of reliable cells will mostly be the same across trials, whereas unreliable cells in one trial may differ from unreliable cells of other trials. Indeed, when examined over all trials of each odor (Fig 1D), a total of 29% of the 124 KCs were unreliable, whereas 6% of the 124 KCs were reliable, which, as expected, is similar to the 5% fraction on an average trial. Fig 1B provides a visual illustration of this reliable-unreliable cell dichotomy. There is only 1 unreliable cell per trial (e.g., cell 4 in trial 1), but there are a total of 3 unreliable cells across all trials (cells 3–5). Thus, on an average single trial, 1 out of 5 cells (20%) are unreliable, and across all trials, 3 out of 5 cells (60%) are unreliable. On the other hand, on an average single trial, 1.5 cells are reliable, and across all trials, 2 out of 5 cells are reliable.

Reliable and unreliable cells also differed in the amplitude of their responses. As the reliability of individual KCs increased, their mean response levels increased exponentially (Fig 1E; panel A in Fig A in S1 Text shows fitted data; panel C in Fig A in S1 Text shows each KC plotted individually and demonstrates a similar trend).

Thus, average measures of activity do not capture the diversity of cellular response properties in third-order olfactory neurons. Responses on individual odor trials are composed of a core set of reliable KCs (about 5% of the population) as well as a peripheral cohort of unreliable cells (about 7% of the population in a single trial, 29% of the population across trials). We next asked if a similar stochastic code is observed in the mouse analog of MB: the PCx.

**Mouse.** We collected new mouse data by imaging responses of 285 PCx cells to a diverse set of 10 odors with 8 trials per odor (Methods, Data Analysis, Fig 8).

The responses of PCx cells bore close resemblance to KC responses in 3 ways. First, a small fraction (3.3%) of PCx cells responded reliably in each odor trial, with a larger fraction (9%) of unreliable cells (Fig 1F). Second, the percentage of reliable cells across all trials was 4.4%, which is close to the percent of reliable cells in an average single trial; in contrast, the percentage of unreliable cells across all trials was much higher: 43% (Fig 1G). It is possible, as we show later, that the increase in the number of unreliable cells in the mouse compared to the fly is due to the increase in the number of trials analyzed (6 trials per odor in the fly versus 8 trials per odor in the mouse). Third, as the number of trials in which a PCx cell responds increases, the size of its response increased (Fig 1H and panel B in Fig A in S1 Text shows fitted data). These results were repeatable for 4 other mice (Methods, Table E in S1 Text).

Thus, odor responses of cells in MB and PCx are similar. The responsive population contains a small number of cells that respond reliably across trials with large responses, while a larger number of cells respond unreliably or infrequently with smaller responses.

## Cell reliability, response size, and odor specificity levels lie on a continuum

As mentioned earlier, splitting cells into 2 classes was a useful simplification and enabled comparison to earlier work [6,8,9]. It is, however, somewhat arbitrary. Indeed, there are gradual differences in response sizes between cells based on their reliability (Fig 1E and 1H). We next

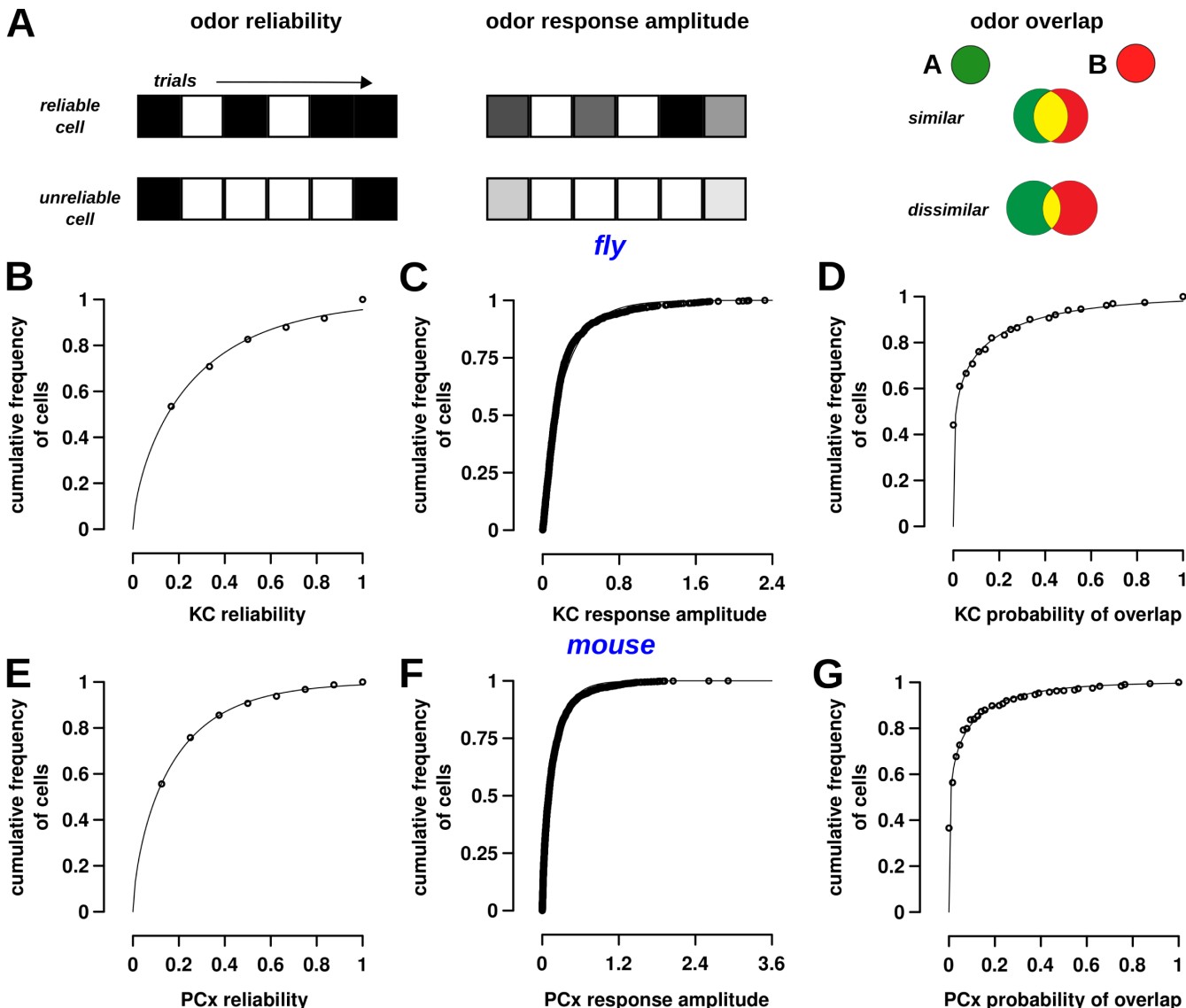

**Fig 2. Cell reliability, response size, and odor specificity follow a continuous distribution.** (A) Schematic of 3 response properties that were examined for odors and odor-pairs. Odor reliability refers to the number of trials for which a cell is responsive. Each trial is depicted by a grid square, and black color indicates a response. Cells that respond in more than half the trials were classified as reliable; all other responsive cells were unreliable. Odor response size is the firing rate of the cell in individual trials, as measured by Ca$^{2+}$ fluorescence. The top line shows a cell with a large response, where the intensity of gray denotes size of response. Odor overlap is the probability that the cell will respond to both odors. The odor overlap here is shown in yellow and decreases for dissimilar odors (bottom) compared to similar odors (top). (B, E) Cumulative frequency histograms showing the distribution of reliabilities for KCs (flies) and PCx cells (mice). Each circle represents the cumulative probability for cells with a reliability value represented on the x-axis. The points are fit by Gamma distributions with shape = 0.64, scale = 0.42 (for B, fly) and shape = 0.64, scale = 0.17 (for E, mice). The significance of the Gamma distribution is that it is a maximum entropy code and optimizes for the most stimuli that can be encoded (see text). (C, F) Cumulative frequency histograms for response sizes in both flies and mice are well fit by Gamma distributions with shape = 0.77, scale = 0.28 (for C, fly) and shape = 0.70, scale = 0.24 (for F, mice). (D, G) Cumulative frequency histograms for the overlap of cells between odor-pairs in both flies and mice, also fit using Gamma distributions with shape = 0.19, scale = 0.60 (for D, fly) and shape = 0.18, scale = 0.35 (for G, mice). Datasets used in these plots for flies (B–D) and mice (E–G) are stored within the Zenodo and Dandi repositories as the main dataset for flies and dataset 164 for mice, respectively. The data underlying the graphs shown in the figure can be found in S2 Data. KC, Kenyon cell; PCx, piriform cortex.

sought to determine whether 3 properties of cells—cell reliability, response size, and odor overlap (described below) (Fig 2A)—fall on a continuum and if there is some structure to the continuum.

The distribution of reliability values of KCs and PCx cells was well-fit to a Gamma distribution in both flies (Fig 2B) and mice (Fig 2E). The parameters of the 2 Gamma distributions were almost identical; fly: shape = 0.64, scale = 0.42; mice: shape = 0.64, scale = 0.28. Most cells have a low reliability—in Fig 2B, the lowest reliability plotted (0.167) starts at a cumulative frequency of 0.5—whereas a few cells have high reliability (the upper part of the curve), reflecting our previous results that there are more unreliable cells than reliable cells. Thus, reliability is a property that changes gradually and is not bimodal with 2 mutually exclusive classes.

The distribution of response sizes of KCs and PCx cells also fit closely to a Gamma distribution in both flies (Fig 2C) and mice (Fig 2F). Again, the fitted Gamma distributions were similar; fly: shape = 0.77, scale = 0.28; mouse: shape = 0.70, scale = 0.24. This result also confirms previous studies [41,42], who showed that olfactory neuron responses follow a maximum entropy code [53]. A code is called maximum entropy if it might encode the maximum number of stimuli using a given population of neurons and set of statistical constraints. Without any knowledge of the statistics of the population responses, the maximum entropy code is a uniform distribution. For example, consider a neuron with a firing rate range of 0 to 100 spikes/s, and it has to encode odor concentration in the range of 0 to 100 ppm. The neuron's coding would be inefficient if concentrations 0 to 80 ppm were represented by firing in the range of 0 to 50 spikes/s and 81 to 100 ppm by 51 to 100 spikes/s. The most efficient coding system would be to uniformly map concentration to response levels (0–100 ppm to 0–100 spikes/s) [54]. If, however, the mean of the population (mean response rate of an odor across all cells or of a cell across all odors) is constrained to be the same, the maximum entropy code follows an exponential distribution of firing rates. Analysis of responses in the first 2 stages of the fly circuit [41,42] showed that each odor's response follows an exponential distribution such that only a few cells are highly active for each odor, and the mean activity rates was the same. The study predicted that the distribution of responses in the third stage (Kenyon cells) would be a Gamma distribution, because each KC integrates 6 to 8 project neuron inputs, and a convolution of exponentials is a Gamma distribution. Incidentally, a Gamma distribution is also a maximum entropy code, wherein the mean and mean of the log values of the population are constrained. Our results show that MB and PCx responses indeed follow a Gamma distribution and optimally encode information. Thus, reliability and response level properties of cells in MB and PCx change gradually.

We developed a third property, which we call overlap, to capture the degree of similarity in cells' responses to pairs of odors. For each cell, we computed the probability that the cell will respond to both odors by multiplying its response probability to each odor (response reliability is converted to probability by normalizing to number of trials). For example, if a cell responds to the first odor in 4 out of 6 trials, and to the second odor in 2 out of 6 trials, the probability of responding to both odors is 8/36, and the overlap score is 0.22 (panel C in Fig D in S1 Text for an example odor-pair). We repeated this procedure for every responsive cell across all odor-pairs. Then, we calculated the mean overlap value across all cells for each pair of odors. We found that the distribution of overlap values of KCs and PCx cells to pairs of odors also followed a Gamma distribution in both flies (Fig 2D) and mice (Fig 2G), with similar parameters: fly: shape = 0.19, scale = 0.6; mouse: shape = 0.18, scale = 0.35 (panels A and B in Fig D in S1 Text for all odor-pair points plotted by reliability). The distribution implies that 80% of cells have overlap < 0.2 and 40% of cells are grouped at the lowest overlap value. Bias in odor selection is unlikely to drive these relationships since the similarity between the pairs of odors (measured as the correlation between KC representations of odors) is uniformly distributed for flies and mice (Fig E in S1 Text). Thus, Fig 2D and 2G shows that most cells have very little similarity across odor-pairs in terms of their response. This raises 2 questions relevant to discrimination: do cells with high overlap belong to similar odors, and are low overlap cells also

more unreliable? (as Fig 1 showed unreliable cells differ even in between trials of the same odor). We tackle both questions and their implications for discrimination in subsequent sections.

These results highlight 2 characteristics of olfactory responses. First, every odor-cell pair is defined by its response level and reliability. Both characteristics are tightly coupled, such that if a cell has a large response to an odor, it is also likely to respond in most trials. Second, all 3 properties follow a Gamma distribution, which indicates that cells do not fall into 2 simple categories, but rather along a continuum.

The large number of unreliable cells raises the issue that these cells, although significantly responsive for a trial, may respond only in that trial and then stay silent, and, thus, not relevant to behavior or decision-making. In such a case, even if these cells are active for this one trial, they will not influence subsequent behavior or decision-making. We found, however, that this is unlikely. We re-examined fly and mouse datasets (including a fly dataset with 13 trials in Fig F in S1 Text), except this time we considered only the first half of the trials and isolated those cells that responded only once—these would constitute the unreliable cells if the fly experiments included the first 3 trials alone (panels A–C in Fig F in S1 Text). We then calculated their response frequency in the remaining trials (trials 4–6 for flies and trials 4–8 for mice) and found that responses did not match the predicted frequency of 1/3 (panels D–F in Fig F in S1 Text). Instead, the responsive population is a composite of cells with a range of reliabilities, showing that even unreliable cells are likely to respond again, though with varying length of intervals in between, consistent with cell reliabilities following a continuous distribution (Fig 2 shown above). Thus, even cells with low reliability can play a role in subsequent odor learning and behavior.

## Circuit mechanism for producing reliable and unreliable cells

Where does response variability come from? Variability is typically ascribed to sensory noise or noise in inputs to a brain region (e.g., neuromodulators). The inputs to MB are known [55,56], and thus, MB presents an excellent substrate to enquire whether sensory or intrinsic noise generates the response variability observed, or, if other mechanisms are involved.

To explore how noise introduced at different stages in the circuit (Fig 1A) affects the emergence of cells with different reliabilities, we developed a linear rate firing model of the fly olfactory circuit (Fig 3A and 3B). In a linear rate firing model, we assume the output of each neuron is a linear function of its inputs. This assumption does well in recapitulating the dynamics of the olfactory circuit network as a whole in addition to simplifying analysis, as previously done [42,57,58]. As described in Methods, the PN→KC connection structure was based on [59,60] and subsequent analysis of this data by [41]; specifically, each of the 2,000 KCs receives synapses from approximately 6 to 8 of the 50 PN types, with synapse strengths following a Gamma distribution. The distributions of KC→APL and APL→KC synapse strengths were based on [55,56,61] (see Methods for statistics). For other parameters whose distributions were unknown, we explored a plausible range, as described below.

We introduced noise in firing rates of cells (PNs, KCs, and APL) and in synaptic transmission between PN→KCs, KCs→APL, and APL→KCs. Thus, we tested 6 manipulations. For each manipulation, we added multiplicative noise by sampling from a Gaussian distribution with a mean of 0 and an SD in the range of 0% to 150%. For example, to add 15% firing rate noise to a PN with response amplitude, $P_{\text{resp}}$, we first computed the noise as $P_\eta = \mathcal{N}(0, 0.15)$, and then computed the noisy response as $P_{\text{resp}}(1 + P_\eta)$. The noise component for each response was generated afresh for each trial. Our goal was to determine which of the 6 manipulations, under "reasonable levels" of noise, could generate a stochastic code (i.e., more or less

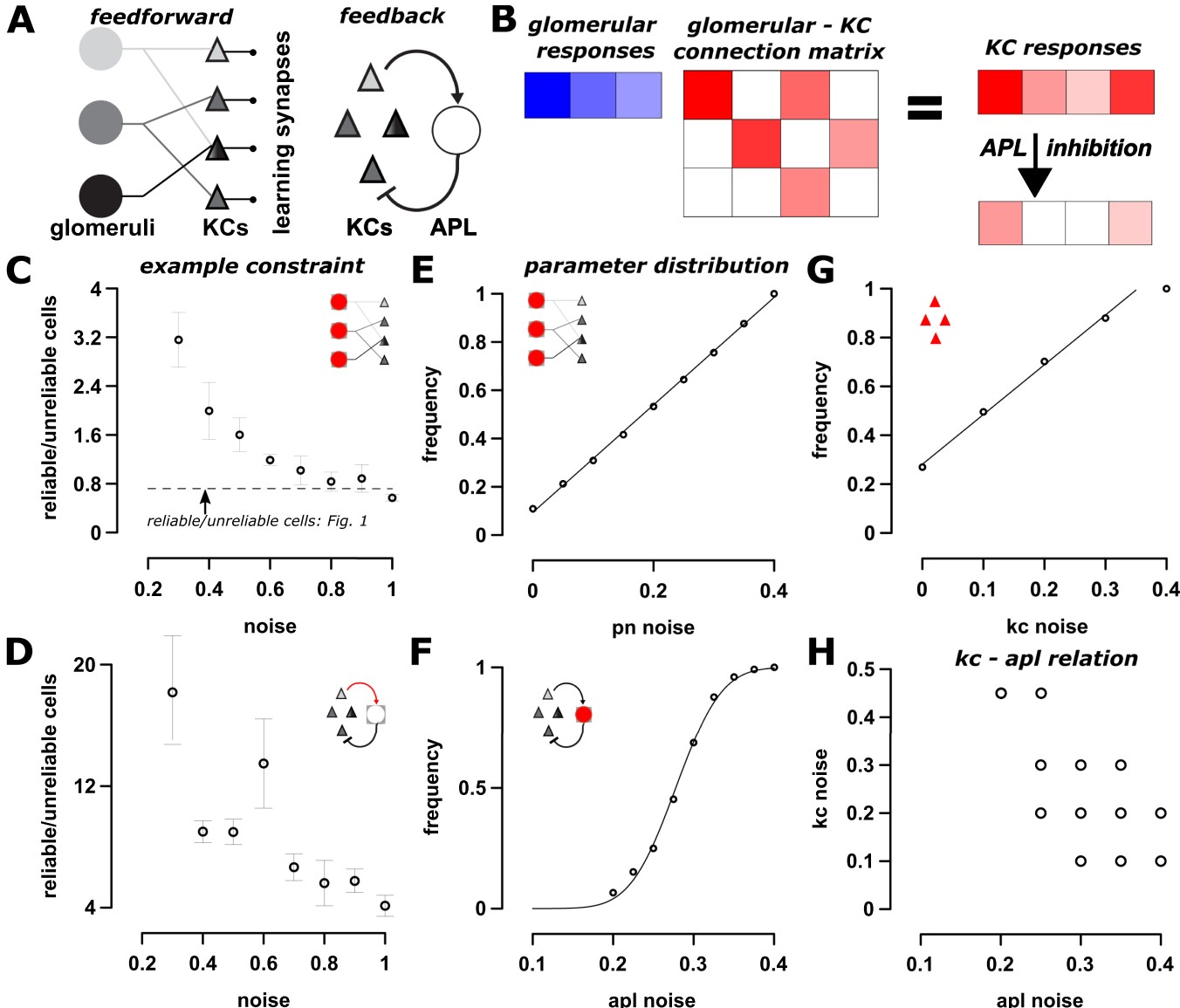

**Fig 3. A WTA mechanism is needed for generating stochastic codes.** (A, left) Schematic of information transfer in the fly olfactory circuit. Odor information from OSNs is passed from glomeruli to KCs (via PNs) in the MB. KCs synapse with MBONs that influence behavior. The KC→MBON synapses are subject to synaptic plasticity. (A, right) The WTA circuit in MB, where all KCs activate the inhibitory APL neuron that in turn feeds back and inhibits all KCs. (B) A schematic of the linear rate firing model. The PN response to odors is depicted in blue, and their connection matrix with KCs is denoted in red. With a linear rate firing model, we take a product of the odor vector and connection matrix to get the KC response, which is sparsified by the APL neuron inhibitory feedback. (C, D) Examples of how the model was checked against constraints. Model simulations show that injecting noise in (C) PNs or (D) KC-APL synapses produces different ratios of reliable to unreliable cells responding per trial. The straight line in the plots denotes a reliable/unreliable cell ratio of 0.72 observed with experimental MB responses (Fig 1). For both plots, the x-axis denotes the fraction of injected noise. Thus, 1 denotes that noise levels are 100% of signal. (E–G) Distribution of individual parameters that make up the "successful" parameter sets. Here, the 6 parameters were varied over a large range (Methods), and we have plotted individual cumulative distributions to show the dependence of the stochastic code on various parameters. (E) PN noise is uniformly distributed. (F) APL noise is normally distributed with a mean of 0.28. (G) KC noise cumulative distribution, where the initial noise is uniformly distributed. (H) There is a an inverse relationship between APL and KC noise. With increasing KC noise, the amount of APL noise that is required for the stochastic code is reduced. Error bars indicate SEM. See Table B in S1 Text, Fig J in S1 Text, Fig K in S1 Text, Fig G in S1 Text, and Methods for more details. APL, anterior paired lateral; KC, Kenyon cell; MB, mushroom body; MBON, MB output neuron; OSN, olfactory sensory neuron; PN, projection neuron; WTA, winner-ake-all.

reliable cells for an odor). These noise additions could arise here, as it does in many other circuits [62,63], through variations in vesicle release probability or in the number of neurotransmitters or synaptic vesicles [27,64,65]. A reasonable level of noise for PNs would be in the range 0% to 50% [47], and for synaptic transmission, would be in the range of 15% to 25% [64,65].

We examined how multiple noise sources could interact, and systematically searched parameter space by combing through all combinations of PN, PN→KC, KC, APL, APL↔KC noise levels, and APL↔KC gain levels. We reasoned that a plausible noise source should be able to replicate the following 5 characteristics of the olfactory stochastic code shown in Fig 1: (i) a ratio of reliable to unreliable cells of 0.72; (ii) # reliable cells per trial of 5.3; (iii) unreliable cells per trial of 7.2; (iv) reliable cells per odor of 6.1; and (v) unreliable cells per odor of 29. The 5 characteristics were well-matched with observations only when physiologically reasonable levels of noise was introduced in the WTA circuit (Fig 3). As we show in Fig K in S1 Text, "successful" parameter combinations also recapitulated other observed features, including response-reliability relationships (Fig 1E), a Gamma distribution of response magnitudes (Fig 2C), and discrimination characteristics discussed later (Fig 6). Additionally, the results are independent of MB size (i.e., number of KCs, panels A–D in Fig K in S1 Text).

To probe how variability emerges within the WTA circuit, we explored 2 models that are distinguished by the connectivity characteristics of the WTA mechanism. In the simpler model, called the single synapse model, there is 1 synapse from APL feeding back to every KC. Briefly, noise in the range of 0% to 100% was added to all components of the model, and we selected combinations that satisfied all 5 characteristics of the stochastic code mentioned above. See Table B in S1 Text and Fig G in S1 Text and Methods ("Modeling and Theory: Noise, Parameter Exploration") for a detailed description of parameter explorations. Of the 44,217 combinations tested, only 2,500 satisfied all 5 characteristics mentioned above, and the top 700 of them had noise in APL→KC connections in the 15% to 30% range. Noise in the other components by themselves did not satisfy all characteristics, highlighting the importance of APL→KC noise towards generating the stochastic code.

The second model, called the multi-synapse model, incorporates recent work highlighting 2 complexities absent in the single synapse model. First, there are multiple synapses between APL and each KC [55,61]; moreover, the number of APL→KC synapses is correlated with the number of KC→APL synapses for each KC, and falls within a range (2 to 38 synapses, Fig 1 of [61,66]). Second, we included noise within KC processes that integrate inputs and push the membrane potential past the spike threshold [67]. A systematic search through this parameter space showed that every successful parameter combination had noise in APL (panel C in Fig J in S1 Text), and the noise fell within a small range that was normally distributed (Fig 3). Moreover, there were some successful combinations that included no noise in PNs, PN→KCs, APL→KC, and KCs (panel C in Fig J in S1 Text). Lastly, while a wide range of noise in feedforward circuit parameters, in conjunction with noise in other parameters, suffice to derive a successful model (Fig 3E), only a narrow range of WTA noise leads to successful models (Figs 3F, Fig J in S1 Text), again indicating the importance of noise in the WTA mechanism for generating stochastic codes.

Why is noise in the WTA circuit crucial for generating stochastic KC responses? A simple explanation is that the nervous system uses averaging to filter out noise from independent sources [27,63]. For example, when synapses from multiple PNs converge onto the same KC, noise from each PN is independent of the others, and collective input noise gets filtered. Conversely, when multiple synapses from a source neuron converge onto a target neuron (e.g., APL to KC), there are 2 sources of noise: synaptic noise that gets averaged and source noise from the neuron itself, which is correlated across synapses and does not get filtered out. In the fly network, noise from the feedforward PN→KC synapses is averaged, as is feedback from the

APL→KC synapses; however, source noise from APL in the APL→KC circuit is not averaged out; rather, it will be added to or subtracted from all synapses. These insights are reflected when considering the effects of noise at each component independently (Figs 3C–3G, panel A in Fig J in S1 Text). For the feedforward circuit (Fig 3A), a biologically unrealistic amount of noise is required (around 80% to 100% for PN noise), while the requirements for noise in the WTA circuit are more plausible (Fig 3F).

What role does intrinsic KC variability play in generating stochastic codes? We found that increasing KC firing noise reduces the amount of noise needed in APL to achieve the observed stochastic code. This is likely because these 2 noise sources are related; i.e., APL activity is proportional to summed KC activity. The summing of 2 related sources of noise is not averaged out, producing the inverse relationship observed between KC noise and APL noise (Fig 3H). Thus, WTA noise might not work in isolation but in conjunction with noise from KCs (e.g., integration noise) or PN noise, to bring it within a biologically plausible range. Additionally, we found that narrower distributions of KC firing rates reduce the WTA noise needed (panel J in Fig G in S1 Text).

In conclusion, our model shows that the WTA mechanism is the best candidate for generating stochastic KC codes.

## Reliable and unreliable cells encode different amounts of odor information

### Preservation of odor similarity from circuit stage 1 to circuit stage 3 olfactory cells.
Are reliable or unreliable cells better at preserving odor similarity as it transfers from the antennal lobe to MB? We compared the similarity of population response patterns in OSNs with the similarity of KC responses across a series of odor pairs. We then split KCs into reliable and unreliable classes and tested how well each class preserved odor similarity. For the former, for odor pair (A, B), we took the set of reliable KCs for odor A and computed their correlation with responses to odor B. Importantly, if a KC is reliable for odor A, it may be reliable or unreliable for odor B, and thus this similarity measure is not symmetric. In panels B and C from Fig 4, there is 1 data point for odor pair (A, B) and another data point for odor pair (B, A). For OSN responses, we used the dataset from Hallem and Carlson [68], who recorded responses of 24 OSN types in the antenna lobe to the same set of odors that were used by [9]. For MB responses, we used the set of 124 KC responses provided by [9].

Consistent with [69,70], we found that most of the odor information in the OSN population is preserved in MB; the Pearson correlation coefficient of odor-pair responses in OSN and KC populations is $r = 0.89$, with a slope of 0.78 (Fig 4A). The slope captures how well the relationship between similar and dissimilar odors is maintained from the antenna to MB; a slope of 1 means perfect maintenance.

When splitting KCs into 2 classes, we found that reliable KCs preserved the similarity of OSN representations and had a large slope ($r = 0.80$, slope = 0.88; Fig 4B) compared to unreliable KCs (Fig 4C), whose range of similarity values was much smaller, likely due to infrequent and weak responses. When odors have dissimilar OSN representations (low correlation), reliable KC population responses also have low correlation, and with increasing OSN representation similarity, reliable KC correlation increases. For unreliable cells, the correlation trend is markedly different. When odors have dissimilar OSN representations, unreliable KC populations have low correlation just like their reliable counterparts. But unlike reliable KCs, as OSN representation similarity increases, there is only a slight increase in unreliable KC correlation, showing that unreliable cells are decorrelated whether odors are similar or dissimilar.

We could not carry out a similar analysis with the mouse dataset, as we did not have odor responses to the same odors in the olfactory epithelium and PCx. We, therefore, compared the

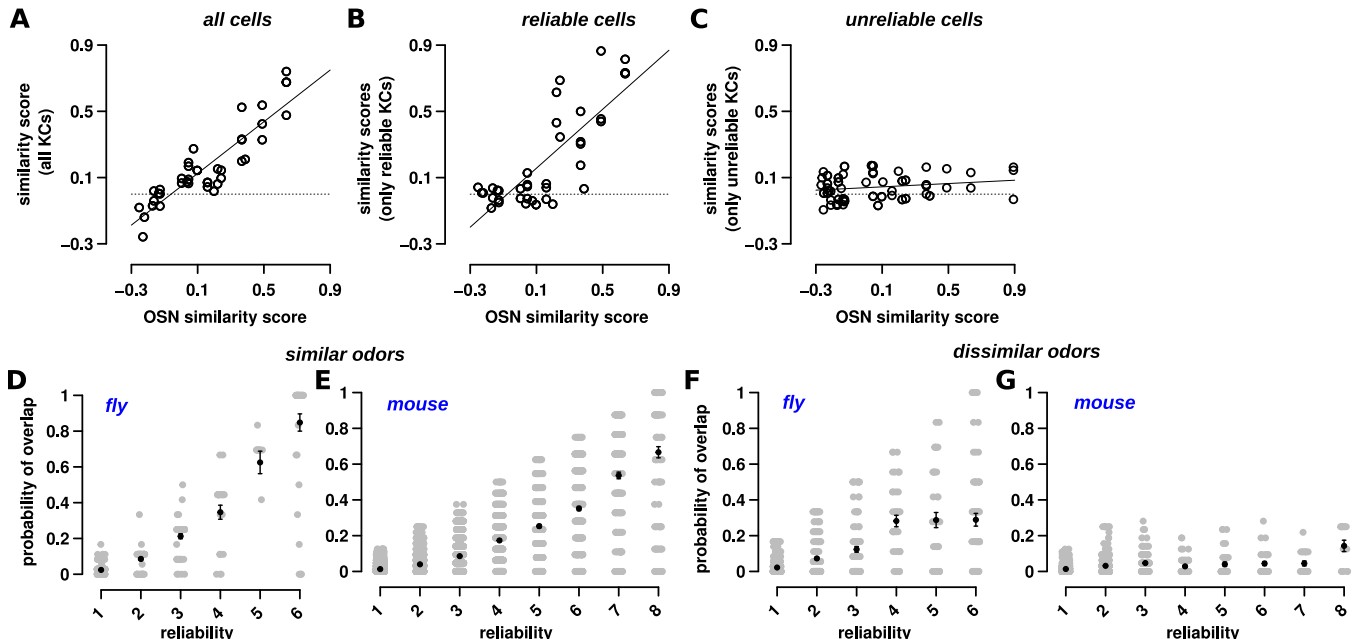

**Fig 4. Reliable and unreliable cells differ in their response to odors.** (A) The similarity between odor pairs was calculated based on OSN response correlation in the fly's antenna [68] (x-axis) and KC response correlation in the fly's MB (y-axis). Each circle denotes 1 odor-pair. Similarity between odor pairs was calculated via Pearson's correlation of response vectors. When considering all KCs, odor similarity was highly correlated ($r = 0.89$). The equation for the line fit is $y = 0.78x$. The dotted line ($y = 0$) in this plot and in B and C indicates the value at which there is no correlation. (B) When considering only reliable KCs, odor similarity is also highly correlated ($r = 0.80$). The equation for the line fit is $y = 0.88x$. (C) In contrast, the similarity between odor-pair responses for only unreliable cells has very low correlation ($r = 0.20$). The line fit has equation $y = 0.05x$, i.e., the slope is close to 0. In other words, odor pairs that are dissimilar (left part of the x-axis) have only slightly less correlation than similar odor pairs (right part of the x-axis). (D, E) In both flies and mice, reliable cells overlap more than unreliable cells for similar odors (i.e., odor-pairs with correlation > 0.5 in their odor responses). Here, overlap is the probability that a cell will respond to both odors, e.g., if the probability of response to the 2 odors is 1/6 and 3/6, the overlap is 3/36 or 1/12. Each gray circle shows the overlap between 2 odors for a single cell, whose reliability for the first of the odor pair is shown on the y-axis. Pearson's correlation between overlap and reliability is 0.87 and 0.91, for flies and mouse, respectively. (F, G) For dissimilar odors (correlation < 0.15), while reliable cells have a higher overlap than unreliable cells, the increase in overlap is much less than for similar odors shown in (D, E). Correlation is 0.55 and 0.33 for linear fits, with slopes of 0.05 and 0.009, respectively, for flies and mice. Error bars indicate SEM. Datasets used in these plots for flies (A–C, D, F) and mice (E, G) are stored within the Zenodo and Dandi repositories as the fly main dataset and dataset 164, respectively (see Data Availability section). Additionally plots A–C used the following supplementary fly datasets: 09042009, 110108, 110109_1, 110109_2. The data underlying the graphs shown in the figure can be found in S3 Data. KC, Kenyon cell; MB, mushroom body; OSN, olfactory sensory neuron.

similarity of odor-pairs with all PCx cells versus just the cohort of reliable or unreliable cells separately (panels E and F in Fig C in S1 Text). For flies, this analysis recapitulated the results of Fig 4A–4C: reliable cell and "all cell" population similarities (of odor-pairs) were proportional while those of "all cells" and unreliable cells were decorrelated. For mouse, we observed a similar trend, suggesting that in PCx cells, too, unreliable cells are decorrelated between all odor-pairs, while reliable cells are decorrelated only among dissimilar odors (with low correlation between odor-pairs).

Next, as an alternative analysis, we used the overlap measure, introduced earlier (Fig 2D and 2G), to capture how a cell's reliability and odor similarity are related. For each set of cells (with the same reliability), we calculated the individual overlap measure of each cell and the average for the whole population. Thus, this overlap measure computes how well reliability and odor similarity are related. We defined similar odors as those odor pairs with a correlation > 0.5 in both MB and PCx (4 similar odor-pairs in flies and 3 similar odor-pairs in mice). Conversely, odor-pairs that had a correlation < 0.15 were designated as dissimilar odors (8 dissimilar odor-pairs in flies and 7 dissimilar odor-pairs in mice).

For similar odors, in both flies and mice, the higher the average reliability of a cell for both odors, the higher the overlap (Fig 4D and 4E). In other words, as reliability (*x*-axis in Fig 4D and 4E) increases, the overlap increases proportionally: fly: $r = 0.87$, mouse: $r = 0.91$. Conversely, for dissimilar odor pairs (Fig 4F and 4G), overlap increases with reliability more gradually: fly: $r = 0.55$, mouse: $r = 0.33$. As above, the slopes of the correlation lines also slightly decreased from reliable cells (fly: 0.15, mouse: 0.10) to unreliable cells (fly: 0.05, mouse: 0.009). The slope indicates the relationship between overlap and reliability: the higher the slope, the higher the dependence between the 2.

We also carried out a third analysis (panels F and G in Fig D in S1 Text) in which we examined how the selectivity of cells for specific odors changed with reliability. We compared how much the cells at each reliability level overlapped versus how much they would overlap if these cells were randomly shuffled. Consistent with Fig 4D–4G, selectivity decreased with reliability, but, nevertheless, was higher than random chance even for cells with the lowest reliability. Thus, when odors are similar, the higher the reliability of the cell, the higher the chance that it will respond similarly to both odors.

**Reliable cells cannot by themselves distinguish similar odors.** Studies have suggested that sparse coding, which restricts odor coding to the top firing neurons, might aid in the discriminating similar odors by reducing the overlap in their representations. Our findings suggest, however, that sparse coding might not reduce overlap, since reliable cells, which tend to have a larger response (Fig 1E and 1H), are highly correlated (Fig 4) for similar odor-pairs. In both the fly and mouse datasets, we found, as expected, that the more reliable a cell, the more likely it is in the top 25% of highest firing neurons (Fig 5B and 5D). Similarly, the more unreliable a cell, the more likely it is in the bottom 25% (Fig 5B and 5D). The top responding cells, in both flies and mice, reflect the same trend as highly reliable cells: as odor similarity increases, so does the similarity of the top responding cells (Fig H in S1 Text).

We delved deeper into the consequences for sparse coding by looking at the cosine similarity between odors (KC response vectors), for 4 subpopulations—the top 25, 50, 75, and 100 percentiles of responding cells—to test how sparsity level affects the overlap in odor representations (Fig 5A and 5C). These are percentiles of only responsive cells for the odor, which means about 30% of the population in flies and about 40% in mice, on average. We use cosine similarity as a proxy for how difficult it may be for downstream read-out mechanisms to distinguish odors (higher cosine similarity→ more difficult discrimination), since it remains unclear how exactly representations are read-out to drive behavior.

Strikingly, we found that the cosine similarity of odors did not decrease from the top 50% of responsive cells to the top 25% of cells for similar odors (top circles), which would be expected under models that assume that sparser codes lead to less representational overlap. For flies and mice (Fig 5A and 5C), the cosine similarity between dissimilar odors (bottom squares) is much less than similar odors. If the circuit were to increase sparseness by simply changing the threshold of responding cells (by increasing inhibition, for example), this would not actually improve discrimination for similar odors, because the cells that are left to respond are reliable cells, which have high overlap and correlation (Fig 4).

The previous analysis compared the overlap between percentiles of responsive cells, but the conclusion is similar when comparing the overlap between quartiles of responsive cells (i.e., instead of the top 0% to 25% versus 0% to 50% we compared 0% to 25% versus 25% to 50%; panel A in Fig H in S1 Text). The top 25% of cells, similar to reliable cells in Fig 4, are good indicators of odor similarity shown by the greater slope: They are more similar when odors are similar and dissimilar when odors are dissimilar. We found similar results on the mouse dataset (panel B in Fig H in S1 Text), suggesting that sparse coding does not aid in pairwise discrimination of very similar odors.

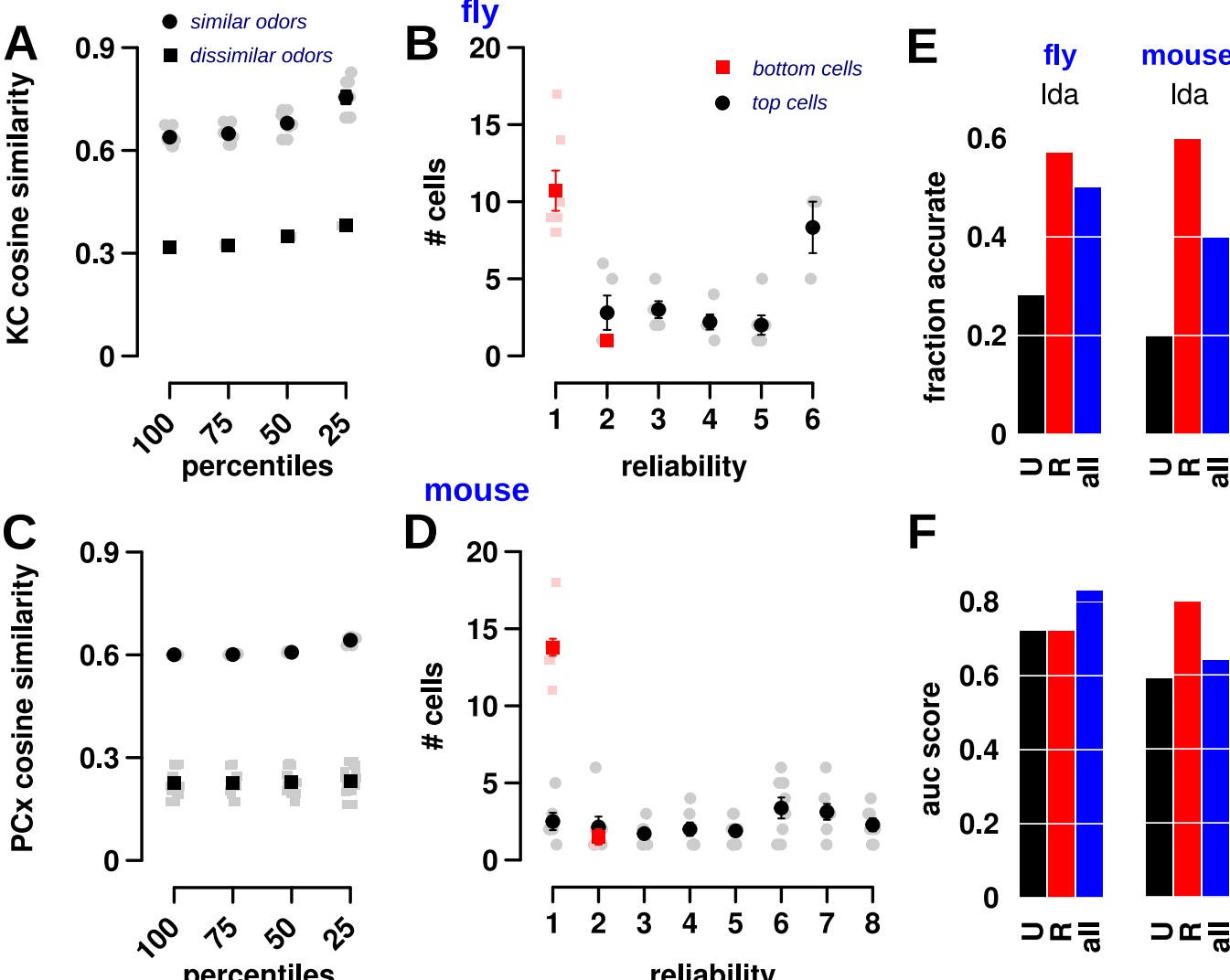

**Fig 5. Reliable and unreliable cells contribute differently to odor discrimination.** (A) The cosine similarity (y-axis) for the top 25, 50, 75, and 100% of responsive KCs (x-axis) for similar odors (correlation > 0.5, top circles) and dissimilar odors (correlation ≤ 0.15, bottom squares). Each square and circle represent odor pairs. The top 25% most responsive cells are more similar than the responses of cells in other percentiles. In other words, the top 25% most responsive cells have the highest overlap for similar odors, and thus they provide less discriminatory power, compared to cells in other percentiles. Cells with high responses demonstrate the lowest overlap for dissimilar odors, but the highest overlap for similar odors. Sparse coding methods typically retain only the highest responding neurons (here, the top 25% of cells). Thus, these results challenge whether sparse coding alone is sufficient for discriminating similar odors. (B) The number of cells at each reliability level for the top 25% (black circle) and bottom 25% (red square) of cells. Most of the top 25% of cells are reliable, while nearly all of the bottom 25% of cells have the lowest reliability. (C) The same plot for mice as (A), showing similar trends. (D) The same plot for mice as (B), showing similar trends. (E) Simple machine learning algorithms (LDA) applied to reliable, unreliable, and all cells show that reliable cells are significantly better at classification than unreliable cells. Compare the length of the red bars (reliable, R) to the black bars (unreliable, UR), where the length of the bar denotes the fraction of correct classifications. The performance of the decoder with all cells (blue bars) is similar to reliable cells, though slightly less. See Methods and Table C in S1 Text for algorithm details and parameters for the LDA decoder and the training and test sets. See panels G and H in Fig C in S1 Text for analogous results with SVM and kNN decoders. (F) AUC (averaged over all pairs of odors) for assessing the pair-wise discrimination performance of unreliable (U, black bars), reliable (R, red bars), and all (blue bars) cells for LDA. Reliable cells, on average, have slightly higher AUC measures. Unlike accuracy measures from (D); however, unreliable cells (in black) show significantly higher AUC, suggesting that they provide significant information for distinguishing odors. See panel D in Fig C in S1 Text for area under the ROC (AUC) measure for assessing performance of the classifiers specifically for similar and dissimilar odors. Datasets used in these plots for flies (A, B, E) and mice (C, D, F) are stored within the Zenodo and Dandi repositories as the fly main dataset and mouse dataset 164, respectively. The data underlying the graphs shown in the figure can be found in S4 Data. AUC, area under the curve; KC, Kenyon cell; kNN, *k* nearest neighbor; LDA, linear discriminant analysis; SVM, support vector machine.

**Discriminating odors using reliable and unreliable cells.** Which cells, reliable or unreliable, are more informative in discerning odors? To answer the question, we used 2 approaches.

In the first approach, we used simple machine learning classifiers on the fly and mouse datasets. The premise is that a decoder might approximate decision-making neurons downstream of MB or PCx that identify odors and tell them apart, and, thus, might serve as a proxy for determining the amount of discriminatory information provided by reliable and unreliable cells. Briefly (see Methods: Data Analysis), for each dataset, we isolated the reliable cells by setting the responses of unreliable cells to 0. We then split all odor trials into training (approximately 80% of trials) and test (approximately 20% of trials) sets. There were a total of 7 odors and 42 trials for flies, and 10 odors and 80 trials for mice. We repeated the same procedure for unreliable cells and all cells together. We applied this procedure to 3 decoder models: linear discriminant analysis (LDA), $k$ nearest neighbors (kNN), and support vector machines (SVMs) with a linear kernel.

Fig 5E and panel G in Fig C in S1 Text show the test accuracy of each decoder in predicting the correct identity of the odor. Reliable cells (red bars; the bar heights represent the fraction of odors that were correctly classified, and the values are presented in Table C in S1 Text) are much better at classification compared to unreliable cells (black bars). Classifying with all cells is significantly better than with unreliable cells, though not as good as reliable cells.

Fig 5F and panel H in Fig C in S1 Text show the area under the curve (AUC) results of each decoder for the test set when telling apart pairs of odors. Given 2 odors A and B, AUC is the probability that a random trial of odor A has a lower chance of being classified as odor B, than as odor A. Thus, a score of 1 shows that A can be easily distinguished from B, while 0 indicates a high likelihood of confusing the 2 odors. In general, odor-pairs that had low AUC had highly correlated KC representations (panel D in Fig C in S1 Text similar, $r \geq 0.5$, AUC: 0.026), and odor pairs with high AUC had decorrelated KC representations (panel D in Fig C in S1 Text dissimilar, $r < 0.25$, AUC: 0.95). Fig 5F shows that reliable cells still provide, on average, a high level of distinguishing information (plotted values are presented in Table C in S1 Text). What is also evident, however, is that unreliable cells, too, provide significant distinguishing information between any pair of odors. Thus, our analysis of the data using linear (and nonparametric) decoders and statistical analysis shows that discrimination between odors becomes hard when their similarity increases, reflecting experimental findings of difficulties organisms face when discerning similar odors [3,9,71,72].

**The role of unreliable cells towards discriminating similar odors.** So, what value might unreliable cells provide for distinguishing odors? Fig 5 shows that in some cases, reliable cells alone perform better at discrimination than all cells; however, Fig 4D and 4E shows that for similar odors, unreliable cells are less overlapping than reliable cells. How might the circuit take advantage of this non-overlap provided by unreliable cells to better distinguish similar odors?

To understand the contribution of unreliable cells, we used a learning model similar to the fly learning and discrimination system [31,73,74] (Fig 7A), since its architecture is better mapped than that of the mouse olfactory system. Learning in the fly system occurs by coordination among 3 sets of neurons: KCs that encode odor information, dopaminergic neurons or DANs that convey reinforcement signals (i.e., reward or punishment), and MB output neurons or MBONs that drive approach or avoidance behavior [31]. During appetitive training, DANs for reward depress synapses between KCs active for the odor (conditioned stimulus or CS) and MBONs that drive avoidance. This tilts the balance such that, when an odor is presented, approach MBONs respond at a higher rate than avoid MBONs, driving approach behavior (Fig 7A→7B, initial training). Avoidance behavior operates in the opposite way by weakening

synapses between CS-responsive KCs to approach MBON. The actual mechanism for discrimination given KC synaptic weights to avoid and approach MBONs remains to be demonstrated —e.g., is it subtractive? Divisive? We simply assume that the larger the difference in the weights, the easier it is to discriminate, but other mechanisms, e.g., division, may exist as well. With this model, we asked, given a test trial, what are the contributions of reliable and unreliable KCs towards overall discrimination?

We illustrate the answer by considering the case where odor A is trained with a reward and odor B is trained with punishment, and pick an example KC x, which responds to A in 2 trials (out of 6 total trials) and to B in 4 trials. Thus, x's response probability is 1/3 for A and 2/3 for B.

$$D_x(A, B) = Pr[x \text{ responds to } A](w_{x \rightarrow \text{approach}} - w_{x \rightarrow \text{avoid}})$$

$$= (1/3)(2 - 4) = -2/3 \tag{1}$$

$$D(A, B) = \sum_x D_x(A, B) \tag{2}$$

There are 3 equations that effectively describe the contribution of KCs towards discrimination, when A is presented in a test trial. Intuitively, the contribution of x depends on how its response differs between both odors. Eq 1 describes the contributions ($D_x$) of x towards approach. Eq 2 describes the total discriminability of odor A from odor B, summed over all KCs x. The first factor (1/3) in Eq 1 represents the probability of activating cell x when A is presented. The second factor represents the difference in the synaptic weights ($w_x$'s) of the connections from x to approach and avoid MBONs.

The $w_x$'s are determined by:

$$w_{x \rightarrow \text{approach}}(t) = \max(w_{x \rightarrow \text{approach}}(t - 1) - \delta \times r_x, 0) \tag{3}$$

Eq 3 outlines how the weights $w_{x \rightarrow \text{approach}}$ and $w_{x \rightarrow \text{avoid}}$ (initially high and equal) change with each training trial. Consider exposure to A and reward: $w_{x \rightarrow \text{avoid}}$ decreases in proportion to the size of x's response ($r_x$) and the learning rate $\delta$. The same rules apply to exposure to B and punishment and $w_{x \rightarrow \text{approach}}$. The t variable represents trial number or time, and shows that $w_x$ will be updated differently depending on whether it is a reliable or unreliable KC synapse. The depression in synaptic weights is faster for reliable cells as they are active more often (83% of the time, Fig 1) with larger responses. Moreover, as synapses cannot decrease beyond 0 (0 term in Eq 3, saturation of synapses), reliable KC synapses will approach 0 quickly, i.e., after very few trials. By contrast, unreliable cell synapses will require many trials to reach 0 weight as they are active less often (25% of the time, Fig 1) with smaller responses.

The $w_x$'s (of reliable and unreliable cells) also depend on the amount of training they undergo. As per [71], normal training is the shorter training period required for discriminating dissimilar odors (e.g., 6 trials for flies [9,75]), and extended training is the longer training required to distinguish similar odors (e.g., 100 trials). Revisiting Eq 1, let us assess the effect of a normal training regimen. $D_x$ depends on whether x is a reliable, unreliable, or silent for odors A and B. Table A in S1 Text lists the possibilities. $D_x$ is large when x is reliable for one odor but not the other. With extended training, if x is responsive to both odors (whether as reliable or unreliable), $D_x$ will be close to 0 as reliable and unreliable synapses will have saturated. Thus, $D_x$ is large only when x is silent for B. Since there are many more unreliable cells than reliable cells and they are unlikely to overlap for 2 odors, the overall discrimination increases with extended training.

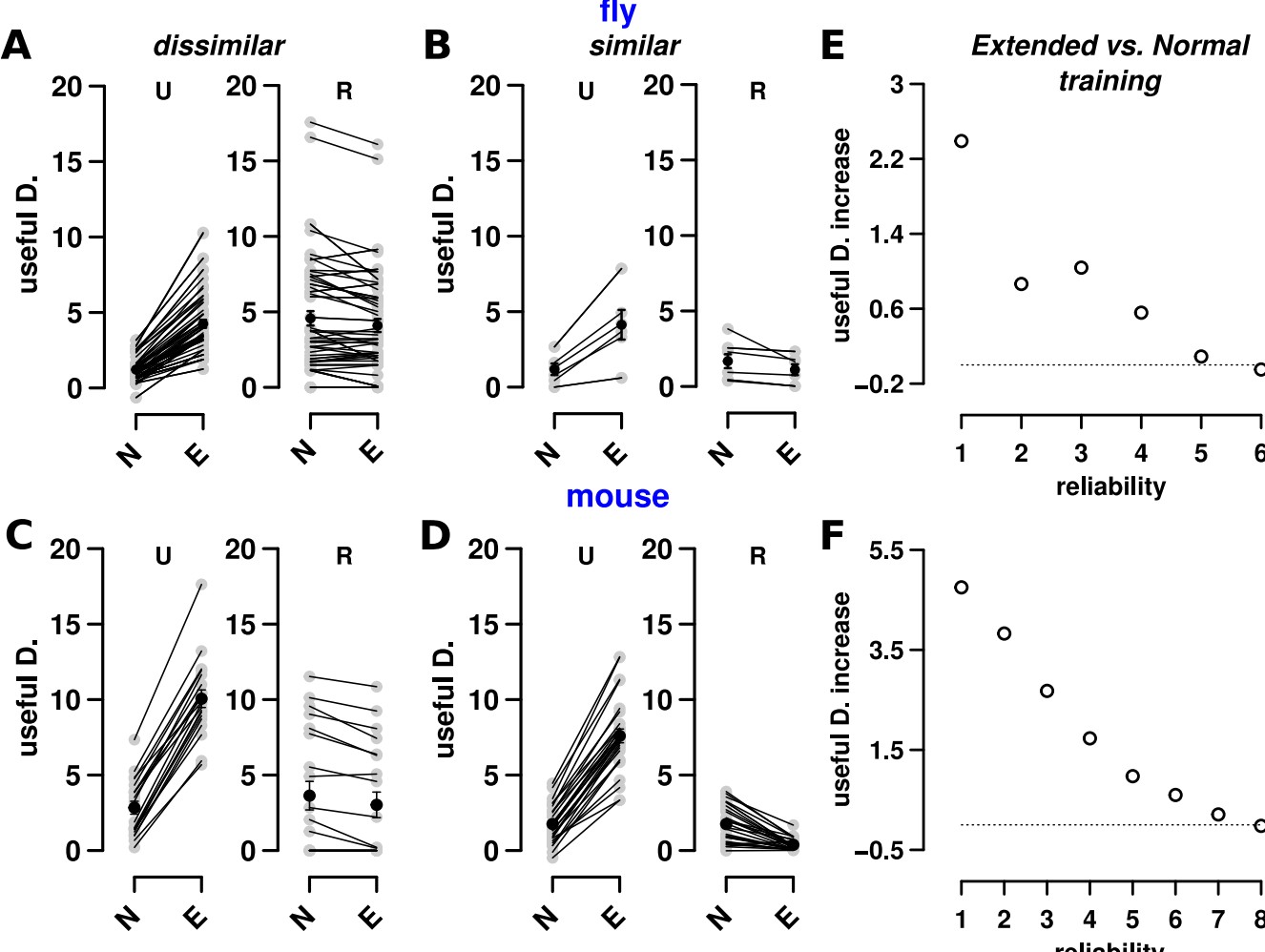

Legend: D. - discrimination, N - Normal training, E - Extended training, U - Unreliable cells, R - Reliable Cells

**Fig 6. Unreliable cells contribute more to discrimination of similar odors.** Extended learning increases the amount of discriminatory power between odor-pairs. Odor-pairs are split into dissimilar (A, C) odor-pairs, whose correlation coefficient is $\leq <0.15$, and similar (B, D) odor-pairs with a correlation $> 0.5$. (A) Shown are 2 plots of how extended (E, x-axis) and normal (N, x-axis) training affect the contributions of (y-axis) unreliable cells (left plot, UR) and reliable cells (right plot, R) towards discrimination (termed useful Discrimination) of dissimilar odors. Each gray point represents the discrimination measure (Eq 2) for an odor-pair. Discrimination calculations for the fly are based on data from [9]. Unreliable cell contributions increase with extended training, to match the contributions of reliable cells. (B) Similar plots as (A) for similar odors. Here, too, extended training increases the discrimination contribution of unreliable (UR) cells. For reliable cells, however, contributions are maintained or reduced; (C, D) are equivalent calculations for the mouse dataset shown in Fig 1. (E, F) The improvement in discrimination from normal to extended training for similar odor-pairs in flies (E) and mice (F). Each circle represents the contribution of cells in a reliability class (x-axis) towards the ability to discriminate similar odors with extended training compared to normal training (y-axis). The highest contribution towards discrimination comes from cells with a reliability of 1, and the contributions decrease as the reliability increases. The dotted lines ($y = 0$) indicate the value at which there is no change in useful discrimination. Datasets used in these plots for flies (A, B, E) and mice (C, D, F) are stored within the Zenodo and Dandi repositories as the fly main dataset and mouse dataset 164 (links in Data Availability section). Additionally, plots A–C used the following supplementary fly datasets: 09042009, 110106_4, 110107_2, 110109_1, and 110109_2. The data underlying the graphs shown in the figure can be found in S5 Data.

The results of applying MB/PCx responses from Fig 1 to Eqs 1–3 are shown in Fig 6A–6D. Three features become apparent for both species. First, with (N)ormal training, the contribution ($D_x$) of unreliable cells ($x$) is about the same for all odor-pairs, while reliable cell contributions decreases from dissimilar to similar, because reliable cells for similar odors are highly likely to overlap. Second, as reliable cell synapses are already saturated with (N)ormal training,

their contributions do not increase for (E)xtended training. Some reliable cells even have reduced contributions—as shown in row 2 of Table A in S1 Text—with extended training, when their response to odor B is unreliable. Third, unreliable cell contributions increase with extended training.

Thus, with normal training, the discrimination measure for the whole population is reduced from dissimilar to similar odor pairs because of the reduced contributions of reliable cells owing to the correlation in their responses to similar odors. With extended training, however, the discrimination measure for the population goes up again as the contribution of unreliable cells (which are decorrelated irrespective of odor similarity) goes up. This effect becomes clear when we consider the contributions of cells from each reliability class separately in extended versus normal training (Fig 6E and 6F). As shown earlier, cells with lower reliabilities are more numerous than those with higher reliabilities (panel D and E in Fig D in S1 Text). Additionally, cells with lower reliabilities are unlikely to respond the same way to 2 similar odors (Fig 4). As a result, as reliability decreases, the collective contributions of cells increases (Fig 6E and 6F), resulting in less reliable cells playing an important role in discriminating similar odors.

These analyses show that reliable and unreliable cells may encode different kinds of information, which are both useful towards identifying odors. As an illustration, let us say the target odor (*A*) belongs to the citrus family of fruits, and the comparison odors (*B*'s) all belong to the berry family. In this case, reliable cells provide distinguishing information, because, for dissimilar odors, reliable KCs are decorrelated. On the other hand, if the comparison odors also belong to the citrus family, then reliable KCs provide less distinguishing information, since reliable KCs are highly correlated for similar odors. Unreliable KCs, however, can provide distinguishing information, as they are decorrelated for similar odors. We propose that reliable and unreliable cells may provide different levels of odor identity information, with reliable KCs providing first-order identity information, and unreliable KCs providing second-order identity information (e.g., finer, more subtle features of the odor), which becomes important as odors become similar.

## Discussion

### Summary of results

We showed that higher sensory regions responsible for odor discrimination and learning, in both flies and mammals, encode odors using a stochastic code. The stochastic code comprises cells that fall along a reliability continuum (i.e., the fraction of odor trials for which a cell responds). Highly reliable cells respond in most odor trials with larger responses. By contrast, less reliable cells respond in fewer trials with smaller responses. With a linear firing rate model of the fly olfactory circuit, we show that the major driver of the stochastic code is noise in the WTA circuit between the APL neuron and KCs and is unlikely to be a product of sensory noise alone. Finally, using the fly olfactory association paradigm on response data, we show that such a stochastic code can enhance discrimination ability.

Building on these observations, we: (i) discuss insights into neural circuit mechanisms that give rise to stochastic codes; (ii) hypothesize benefits for stochastic codes and propose models for how reliable and unreliable cells work together to facilitate odor discrimination; (iii) hypothesize possible learning and discrimination mechanisms in flies and mice that leverage stochastic codes; (iv) revisit the role of sparse coding for fine-grained discrimination; and (v) describe other neural circuits and brain regions where benefits of trial-to-trial variability have been appreciated.

## Neural circuit mechanisms for generating stochastic codes

There are 2 inputs that shape the activity of an odor-coding cell in MB: inputs from PNs and from APL. First, each KC samples from approximately 6 of the 50 PN types, and this input plays a role in whether the cell is reliable or unreliable. For example, a KC is very likely to be reliable for an odor if most of the 6 PNs it samples from are highly active (large response) for the odor. This would allow the KC to survive any reasonable amount of noise or variance in the WTA threshold. On the other hand, a KC is more likely to be unreliable if only half of the PNs it samples from are highly active. Combinatorially, there are many distinct odors for which only 3 or fewer out of 6 PNs are highly active, but fewer odors where all 6 are active. Similarly, if a specific PN→KC synapse is strong, then perhaps only that 1 PN need be strongly active for an odor for the KC to be reliable, but there are likely many more odors where that PN is mildly active. Thus, for any odor, there are many more unreliable cells than reliable cells.

Considering the stochastic component, we experimented with 3 possible causes of variability with our model. Two of the causes were based on sensory noise arising in the first or second stages of the circuit (i.e., noise in PN firing rates or in PN→KC synaptic transmission). For both, it was implausible to generate the observed stochasticity without physiologically excessive noise levels. Previous studies support this view [70], showing that in the mouse PCx circuit, each neuron receives synapses or inputs from many OB neurons, and noise reductions in one input are likely to be offset by noise additions in other inputs. The fly has fewer inputs into each KC in comparison, but even here, averaging is likely to filter some noise. In addition, connectome analyses of MB have found compensatory variability in the fly circuit; e.g., weights of excitatory PN→KC connections are inversely correlated with the number of PNs each KC samples from [76], effectively canceling out variability, though such remodeling likely occurs at longer time scales than the experiments analyzed here. Thus, for moderate noise regimes, odor signals are robustly transmitted to the third stage of the circuit making it unlikely that the observed coding variability originates from the first 2 stages alone.

The WTA circuit, on the other hand, does not average or cancel out noise at KCs. Each KC gets direct inhibition from APL. Computationally, this implies that in each odor trial, each KC $x$ receives a slightly different amount of inhibition, which alters the relation between its firing rate, $r_x$, and its threshold for activation, $\tau_x$. For example, if KC $x$ has $r_x \approx \tau_x$ without noise, on one trial noise might lead to $r_x < \tau_x$ and the cell staying silent, and on the next noise might lead to $r_x > \tau_x$ and the cell being active. Such cells, whose $r_x$ is close to $\tau_x$ without noise, form the cohort of unreliable KCs. When $r_x \gg \tau_x$, KC $x$ survives inhibition regardless of the noise, and these are the reliable KCs.

Although we show that the WTA circuit is likely a core driver of the observed variability, our results indicate that other components (e.g., KC noise) also contribute, and indeed, studies have shown noise levels in PNs and KCs range up to 50% [47,67]. Further, our results that the WTA circuit is the best candidate to generate KC response variability may need to be revisited if the *Drosophila* connectome reveals other connections, such as KC-to-KC feedback loops, or if the model is applied to the mouse olfactory system, where there are a population of inhibitory neurons that play the role of APL [35].

The mouse PCx, like the fly MB circuit, contains a WTA circuit between PCx principal and inhibitory cells (albeit in a different form [32,49,50]), providing multiple synapses where variability can arise. Variability may also arise through other sub-circuits in PCx. One of them is the excitatory recurrent circuit between PCx principal cells [49], which accounts for nearly half of excitatory PCx principal cell inputs [70,77]. Additionally, OB input into layer 2 and 3 cells is modulated by a feedforward inhibitory circuit between mitral cells in the OB and principal cells in PCx [32,78]. Finally, PCx cell activity is likely influenced by an OB→PCx→OB feedback circuit

that has the characteristics of a WTA circuit, in negatively suppressing its own activity. A portion of PCx principal cells (which are activated by mitral cells) excite granule cells in the bulb that then inhibit mitral cells [79–81], and, could, in turn, change OB input to PCx and PCx activity.

Thus, processes that convey sensory information from the periphery collaborate with inherent circuit mechanisms within MB and PCx to encode odor responses using a stochastic code. The contribution of the WTA circuit, as opposed to input noise, highlights an intriguing possibility of neural circuit design. All circuits might contain mechanisms to filter noise from upstream circuits, while also containing mechanisms that generate inherent noise that aid in better exploration of the environment, or as we suggest, better discrimination.

Finally, our study is compatible with work showing that odor identity is also encoded by a temporal or primacy code [82,83], in which identity is derived from the combination of neurons that respond the earliest [84]. Reliable cells, which respond more frequently with larger responses (spikes/s), would have an earlier response than unreliable cells. Recent findings [50,85] showing that fine-discrimination requires more time, with the PCx WTA circuit facilitating this discrimination, support this view and are consistent with our findings.

## Challenging the role of sparse coding

It has long been argued that increased sparsity in odor representations leads to increased pattern separation between 2 odors [86]. While sparse coding may help discriminate dissimilar odors, sparse coding alone may not be sufficient for fine-grained discrimination of 2 very similar odors. The highest firing cells (i.e., those that would remain firing if odor representations were made sparser by increasing inhibition) would be reliable cells, which have more overlap, not less, between similar odors (Figs 5, Fig H in S1 Text). An intriguing alternative possibility, for flies, is that the inability to discriminate between similar odors arises from the insufficient effect of unreliable cells, as they do not experience enough trials under standard training protocols.

Why, then, is there a need for sparse coding? There are 2 potential uses for it. First, sparse coding is integral to discrimination, not in separating 2 odors, but in separating many odors; i.e., for enhancing the coding capacity of the system. When odor codes are sparse, they use fewer cells, and thus when new (different) odors are introduced, they are less likely to interfere with stored odors. For example, if an odor is encoded by 5% of KCs, and we conservatively assume that new odors can be learned only if they are completely non-overlapping with preexisting odors, then 20 odors can be learned. If the odor code were less sparse, at say 10%, the capacity of the circuit would be lower at 10. While actual discrimination may be more forgiving with respect to odor overlap, the notion that coding capacity is inversely related to sparse code size still applies. Thus, sparse coding may be important for optimizing the coding capacity of the circuit but our results suggest that it may not improve discrimination between similar stimulus pairs. Further, there is a limit to how sparse a code can be. Odor recognition using very sparse codes (e.g., if only 1 or 2 neurons were to encode odors) is likely less robust to sensory noise and other environmental nuisances. Second, while sparse coding might not aid discrimination of similar odors, it reduces the computational power needed for discriminating dissimilar odors, which might have an evolutionary benefit. Thus, the olfactory system may be geared towards quick discrimination using sparse coding (reliable cells) and towards longer discrimination of similar odors by leveraging unreliable cells.

## Benefits of stochastic coding and possible models for fine-grained discrimination

Perceptual learning is the phenomenon where an animal is unable to distinguish 2 similar stimuli but after repeated (passive) exposure or (active) training sessions, acquires the ability to

do so [71,87]. Responding cells for 2 dissimilar odors will be mostly non-overlapping, which means that reliable cells are by themselves likely to be sufficient for distinguishing dissimilar odors. However, given 2 similar odors, we showed that there is stronger correlation in the activities of reliable cells than the activities of unreliable cells (Figs 2 and 4). Reliable cells could be advantageous if the goal of the animal is to generalize behaviors from one odor to another similar odor. On the other hand, unreliable cells may encode important information (e.g., higher order statistics or more subtle features of the odor) that can, with training, help discriminate the odors. Our model (Fig 6) shows that it is difficult to discriminate similar odors with normal training; however, extended training enables better discrimination by recognizing the information encoded within unreliable cells. While the question of how much distinguishing information MB or PCx needs to discriminate 2 odors remains unknown—e.g., is a single non-overlapping cell sufficient?—it is generally accepted that the more non-overlapping the representations of 2 odors, the easier it is to discriminate [86].

Below, we describe 3 possible models for fine discrimination (Fig 7), and for the purpose of illustration, we use the fly olfactory learning system [31]. These models transition from thinking of an odor as a single point in high-dimensional space to thinking of an odor as a cloud in high-dimensional space. Each point in the cloud corresponds to the representation assigned to an odor in a single trial. In addition, these models require extended training, with many more training trials than is typically performed, since unreliable cells are active in ≤50% of trials. Importantly, these are only schematics of plausible models, intentionally simplifying certain details such as firing rates of responding neurons, noise structure, and other possible read-out mechanisms.

To begin, Fig 7A depicts 2 similar odors, A and B. The responsive neurons for A and B fall into 3 clouds. The pink cloud contains the reliable cells, which are highly overlapping since the 2 odors are very similar. The unreliable cells are less overlapping and are depicted by 2 gray clouds for odors A and B. Since the animal has not undergone training, the strength of the connections that drive downstream approach and avoid MBONs are equally matched. Thus, the fly neither approaches nor avoids the odor upon exposure.

1. Saturate the cloud (Fig 7B). The first model performs perceptual learning by saturating synapse strengths between unreliable KCs and MBONs. The results of Fig 6 show this model in action. First, with initial training—e.g., 12 CS-US trials or a similar amount as in [9], which could be considered as a norm for most studies—the strength of synapses from reliable cells (for the 2 odors being compared) to MBONs are depressed. Second, the strength of synapses from an odor's unreliable cells to the MBON are depressed; however, these synapses are not depressed as much as reliable cell synapses, since unreliable cells are activated in fewer trials. For similar odors, the reliable cells are a "wash," providing little information for distinguishing odors, whereas unreliable cells have begun to separate the odors, though they have not responded in enough trials to make a robust behavioral difference. With extended training, the effect of unreliable cell synapses increases to improve discrimination.

2. Resize the cloud (Fig 7C). In the second model, the cells encoding the 2 odors change with training, such that the cloud for odor A is resized to exclude cells that also respond to odor B. The cloud can change size by converting unreliable cells to reliable cells or by turning off reliable cells (since they overlap the most between 2 similar odors).

3. Move the cloud (Fig 7D). The third model moves the clouds apart via the addition of new cells (colored in yellow) that were not previously responsive to the odor. In addition, unreliable cells may become more reliable with extended training. Recent work by [88] demonstrated that repeated training leading to long-term memory can change KC representations

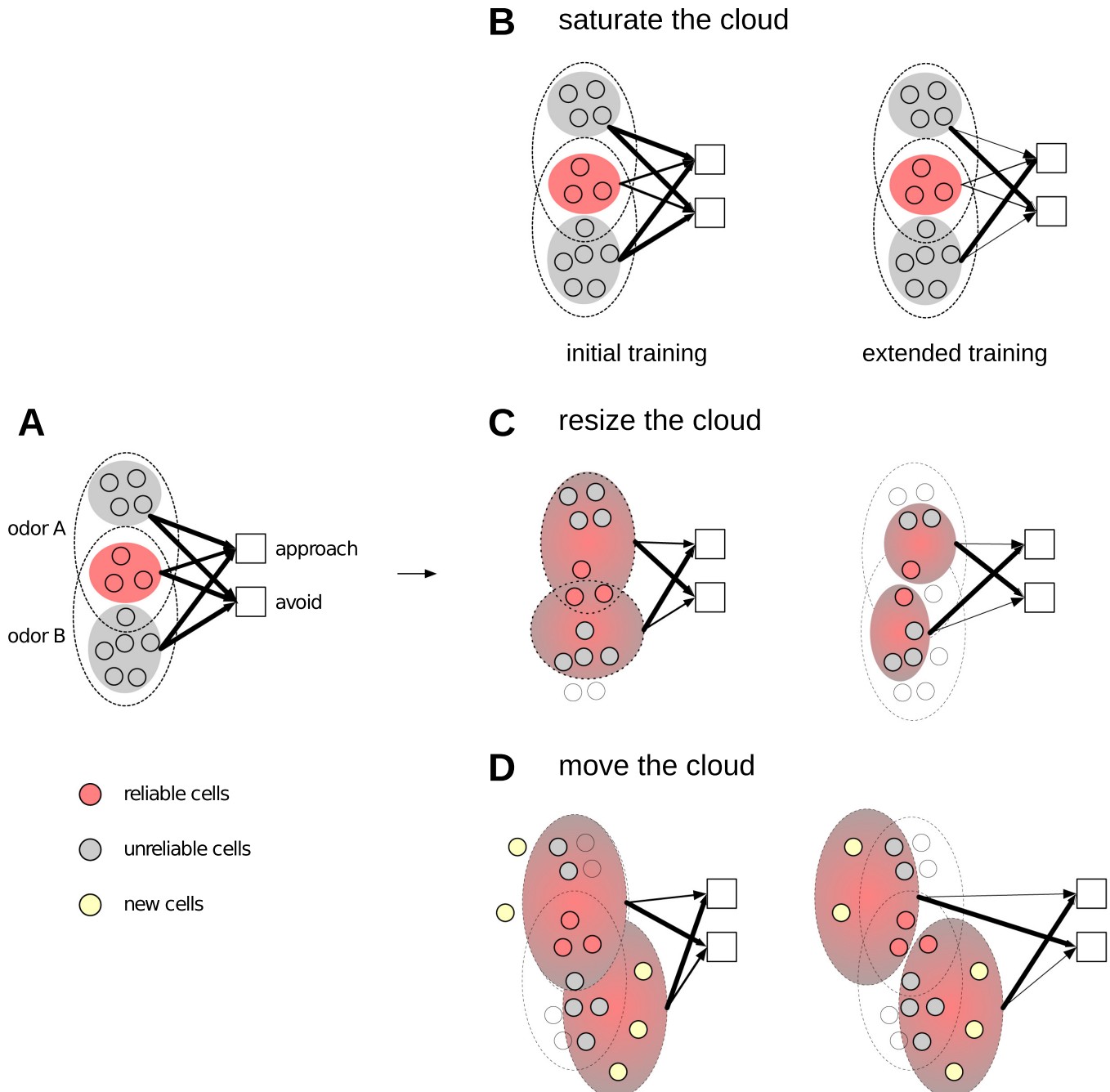

**Fig 7. Three possible models of fine-discrimination.** (A) The stimulus representations in the untrained animal. For each odor, the dotted ovals enclose the cells that respond to the odor (unreliable cells are gray, reliable cells are pink). The unreliable cells are mostly non-overlapping for the 2 odors, whereas the reliable cells overlap exactly. The thickness of the arrows denotes the total strength from the 3 sets of cells to the 2 downstream neurons that drive approach or avoidance behavior. Initially, all arrows are of the same strength. (B) Saturating the cloud. With initial training (left), the strength of downstream synapses of reliable cells saturates (i.e., weakens to near zero), but synapses of unreliable cells barely change their strength. Consequently, as reliable cell overlap is high between the 2 odors, approach/avoid neurons receive equal input, and the animal is unable to discriminate. With extended training (right), the downstream synapses of unreliable cells also saturate, driving the 2 stimuli apart, as depicted by the thicker arrows. The connection strengths to the approach MBON are much weaker than to the avoid MBON, and therefore odor A and B are distinguishable. (C) Resize the cloud. The stimulus representation shrinks to only include cells that are non-overlapping between the 2 stimuli (e.g., the shared reliable cells are turned off). While some resizing may occur with normal training, with extended training, the 2 stimuli are further driven apart, enabling the animal to discriminate the 2 odors. (D) Move the cloud. Some mechanism (most likely driven by unreliable cells, the cells that are most unlike between similar odors) rearranges the stimulus ensemble response to include new cells, or converts unreliable to reliable cells that do not overlap, and thus improve discrimination. MBON, MB output neuron.

in flies. Further, in mice, representational drift [89,90] is another form of "moving the cloud," albeit over longer time scales (days to weeks, as opposed to seconds considering repeated trials in succession), and in the absence of training.

Finally, there remains an important unresolved problem: both saturating the cloud and moving the cloud could potentially cause unintended interference with other odors. For example, when saturating the cloud, unreliable cells for odor A now strongly drive approach behavior; however, these same unreliable cells could be reliable cells for odor C, and if odor C was paired with punishment, there could be a problem. Similarly, when moving the cloud, the new cells recruited to be part of odor A's cloud could previously have been part of another odor's cloud. With all models, although training improves the discriminability of a particular odor-pair, it will affect the discrimination of other odors and the total number of odors that can be distinguished (odor capacity). Thus, in future experimental evaluations of these models, it is important to consider the trade-off between stable odor perception and coding capacity.

## Learning and discrimination mechanisms in piriform cortex and mushroom body

Although, the models sketched here need to be grounded in precise circuit mechanisms, there is evidence for cloud saturation in flies, and evidence of both cloud saturation and cloud moving in mice. In flies, previous experiments that used optogenetic activation of DANs as reinforcement [91] showed that KC representations and amplitudes remain stable with associative training, while MBON activity decreased. While there may be other effects during extended training (e.g., non-associative effects based on anatomical or neuromodulatory feedback from MB to the antenna), these results support the "saturate the cloud" model in flies (Fig 7B).

In vertebrates, there is evidence that cell responses remain stable with learning in mouse [92], while other studies in rat PCx [93], mouse PCx [89], and fish Dorsal Pallium [94] show that they change over time (moving the cloud model). Intriguingly, in a study [89] of representational drift, a small percentage of PCx cells remained stable over time. Based on our findings of 3.3% reliable cells, it is possible that these are reliable cells and might explain the otherwise contradictory results observed; while reliable cells remain more stable, less reliable cells change over time. Even though unreliable cells change, their effect on discrimination would still be compatible with our model (Fig 6) as long as the drift for odor-pairs is decorrelated.

The notion that PCx cells play a part in discrimination comes from an important study examining the discrimination of artificial PCx odors [52], generated by using light on PCx neurons. The ability of PCx ensemble activity (sans sensory activity) to generate discriminatory behavior suggests that learning and discrimination mechanisms are likely in PCx or downstream of it. If they are downstream, in which region could the plasticity mechanisms operate? Several labs have shown that OB responses decorrelate over time, and that PCx seems to be required in the form of PCx → granule cell inhibition of mitral cells [79,81,95,96], with some evidence of plasticity in OB [97]. Notably, OB decorrelation occurs even without reinforcement (which is unlike the fly mechanism). Additionally, if OB encoding changes over time, it might also lead to changes in PCx coding of odors favoring the moving the cloud model. Other candidate downstream areas that might do the work of the KC-MBON-DAN circuit include the Posterior PCx, parts of the prefrontal cortex, or the hippocampal formation [92,98–100].

Thus, learning and discrimination mechanisms in the olfactory circuit could be a hybrid scheme that includes saturation of downstream synapses and changes in odor coding ensembles. We are not aware of cloud resizing in the olfactory system; however, there is evidence of stimulus ensemble resizing in the context of songbird and motor learning, which we discuss below.

## Generality to other neural circuits and brain regions

Trial-to-trail variability in neural responses is common in myriad brain regions and species, and has been observed in both early sensory coding areas, as well as deep layers involved more directly in learning and memory. Other studies have shown at least 3 ways in which variability can improve an animal's fitness.

First, variability expands the global search space. A popular illustration of this phenomenon is in songbirds [25,26,101]. Young zebrafinches learn their courtship song from older adults in a process called directed singing. They also engage in solo practice sessions called undirected singing, where they vary the song. The final song used in courting females is a mix of the stereotyped song that they learn and the variable song that they practiced. Related theoretical studies, especially in computer science, have shown that a crucial part of reinforcement learning algorithms is a random exploration component [102], which helps guide the search away from local minima. The use of variability in song learning is this principle in action. A similar strategy is also observed in motor systems. The amount of learning required is directly proportional to the amount of variability at the start, and as learning or training continues, the amount of variability is reduced [103]. Essentially, the subject explores various possibilities at the start, and then exploits the best solutions found. Here again, variability increases the spread of possible actions and thus the likelihood of hitting a more optimal solution. Thus, for olfaction, it is intriguing to hypothesize that variability in odor coding may vary over time, perhaps tuned by neuromodulators, until some "near-optimal" encoding is found. This notion finds some support in theoretical studies of neuronal networks. Neuronal networks trained in the presence of noise are more robust and explore more states, enabling them to better adapt to dynamically changing environments [104,105].

A second use of variability is the phenomenon of stochastic resonance (SR) found in various species and systems [106–109]. In these systems, the ability of animals to sense objects in their vicinity is determined by the threshold activation of their sensory receptors: the lower the threshold, the farther the object that they sense. At very low thresholds, however, animals might confuse noise and signal. SR allows them to bridge the problem by augmenting high thresholds with random noise events that allow them to sample lower signals, thus allowing detector neurons to sample a wide range, i.e., more information [54,110]. The WTA mechanism is analogous in creating a central-SR process as it modulates the WTA inhibition based threshold of KC activation through noise and enables a wider range of discrimination.

Third, trial-to-trial variability has also been viewed as the brain sampling from a probability distribution of possible outcomes [111,112]. These models require that the animal first learn a generative model of the environment (including a likelihood function and a prior). Stochastic coding may provide a simple implementation of this scheme, where the reliability of a cell can be viewed as the probability that the features encoded by that cell are present in the odor.

In this study, we suggest a fourth, hitherto, unexplored benefit that might accrue from trial-to-trial variability: the ability to improve learning and discrimination, specifically for fine grained discrimination.

Overall, the broad conservation between 2 olfactory circuits that generate stochastic codes suggests that this coding principle, arising through the action of a WTA mechanism within a distributed circuit, might be more universal. Other distributed circuits such as the hippocampus, prefrontal cortex, and the cerebellum have prominent WTA circuits [86], with observed instances of trial-to-trial variability.

## Materials and methods

We split Materials and methods into 3 parts. The first part provides experimental details of the imaging techniques used. The second part outlines the analysis techniques used. The third

describes theory and modeling. For the theory and analysis parts, we used the statistical programming language R [113].

## Experimental methods

**Mice.** Experimental and surgical protocols were performed in accordance with the guide of Care and Use of Laboratory Animals (NIH) and were approved by the Institutional Animal Care and Use Committee (IACUC) at Brown University. C57BL/6J mice were crossed to Ai14 mice [114] and male and female heterozygous transgenic offspring 8 to 12 weeks of age were used. Mice were maintained with unrestricted access to food and water under a 12-h light/ dark cycle and housed individually after surgery.

**Stereotaxic surgery.** Virus (AAV1-syn-jGCaMP7f-WPRE, [115]) was purchased from Addgene and injected using manually controlled pressure injection with a micropipette pulled with a Sutter Micropipette Puller. Mice were anesthetized with Isofluorane with an induction at 3% and maintenance at 1% to 2% with an oxygen flow rate of approximately 1 L/minute and head-fixed in a stereotactic frame (David Kopf, Tujunga, California, United States of America). Eyes were lubricated with an ophthalmic ointment and body temperature was stabilized using a heating pad attached to a temperature controller. Fur was shaved and the incision site sterilized with isopropyl alcohol and betadine solution prior to beginning surgical procedures. A 1.0-mm round craniotomy was made using a dental drill centered to the following stereotaxic coordinates: ML: 3.9, AP: 0.3. Virus diluted to 1/3 in dPBS was injected at 100 nL per minute at 3 different spots (total 1 $\mu$L) using the following coordinates (mm) to target PCx (ML/AP/DV): 3.85/ 0.6/−3.8, 3.95/0.3/−3.9, 4.05/0.0/−4.0, all relative to bregma [116]. After 5 min, the micropipette was slowly retracted from the brain at 500 μm per minute. Following surgery, mice received buprenorphine slow release (0.05 to 0.1 mg/kg) subcutaneously. Lens implantation surgery occurred 2 to 3 weeks following virus injection. A GRIN lens (NEM−060−25−10−920−S−1.5p, GRINTech) was implanted above PCx. The lens was implanted, centered to the craniotomy, at 100 μm per minute until reaching the following coordinate: DV: −3.9. Once placed, the lens was fixed to the skull with Metabond adhesive cement (Parkell, Edgewood, New York, USA). A custom-made aluminium headbar was then attached to the skull using dental cement (Pi−ku−plast HP 36 Precision Pattern Resin, XPdent). Finally, a protective cap over the lens was applied with Kwik−Sil silicone elastomer (World Precision Instruments, Sarasota, Florida, USA). Mice were allowed to recover from lens implant surgery for at least 4 weeks prior to imaging experiments.

**Odor delivery.** Animals were habituated to the experimenter and head-fixation setup for 30 mins a day for at least 2 days before the imaging experiment. On imaging days, odor stimuli were delivered through a custom built 16-channel olfactometer (Automate Scientific, Berkley, CA, USA) equipped with a mass flow controller that maintained air flow at 1 liter per min. The olfactometer solenoids were triggered by a Teensy 3.6. A vacuum was applied inside the two-photon isolation box to evacuate residual odors. For all experiments, mice were habituated to the two-photon head fixed setup for 10 min prior to imaging. An odor trial lasted 30 s (10 s of pre-stimulus baseline, 1 s of stimulation, 19 s of post-stimulus acquisition) with inter-trial intervals of 30 s. Odor stimuli were presented in pseudo-randomized fashion and 8 presentations of each odor were performed in a session. Two odor panel were run as follow (diluted in mineral oil vol/vol): Odor panel 1: benzyl isothiocyanate 10%, cinnamaldehyde 10%, (R)−(−)− Carvone 10%, Dihydrojasmone 10%, Ethylene brassylate 100%, Ethyl decanoate 50%, galaxolide 50%, Isoamyl phenylacetate 10% Odor panel 2: acetophenone 1%, amylamine 1%, butyl acetate 1%, ethyl hexanoate 1%, 2−Isobutyl−3−methoxypyrazine 1%, $\beta$-Ionone 1%, 2,3−Pentanedione 0.1%, Valeric acid 0.1%. A photoionization detector (miniPID 200B, Aurora Scientific, Canada) was used to confirm reliable odor delivery.

**Two-photon microscopy.** A typical imaging experiment lasted approximately 1.5 h per mouse. Two-photon imaging of the PCx was performed using an Ultima Investigator DL laser scanning microscope (Bruker Nano, Middleton, Wisconsin, USA) equipped with an 8 Khz resonance galvanometer and high-speed optics set, dual GaAsP PMTs (Hamamatsu model H10770), and Z–Axis Piezo Drive for multiplane imaging. Approximately 90 to 150 mW of laser power (at 920 nm, from Chameleon Discovery NX Ti:Sapphire laser source (Coherent, Santa Clara, California, USA)) was used during imaging, with adjustments in power levels to accommodate varying signal clarity for each mouse. After focusing on the lens surface, optical viewing was switched to live view thru the two-photon laser, and a field of view (FOV) was located by moving the objective approximately 100 to 500 μm upward. Three FOVs were chosen separated approximately 80 μm apart in depth. Images (Fig 8) were acquired with a Nikon 10× Plan Apochromat Lambda objective (0.45 NA, 4.0 mm WD). GCaMP7f signal was filtered through an ET–GFP (FITC/CY2) filter set. Acquisition speed was 30 Hz for 512 × 512 pixel images. Planes were imaged simultaneously, yielding a final acquisition rate of 4.53 frames per second.

**Imaging data processing.** Suite2p [117] was used for video non-rigid motion correction, cell region of interest (ROI) selection, and calcium trace extraction. Suite2p was run separately for each plane. Putative neurons were identified and sorted by visible inspection for appropriate spatial configuration and $Ca^{2+}$ dynamics. Neuropil signal was subtracted by a coefficient of 0.7.

## Data analysis

Here, we describe the methods used in analyzing the fly and mouse data. In the first subsection, we explain the analysis performed to separate signal from noise and the effect of different *p*-values in selecting the significance threshold for distinguishing signal from noise. In subsequent subsections, we present analysis of the imaging response data. To make it more accessible, we list these subsections by figure.

**Analysis of imaging data.** The odor responses recorded from PCx in mice, and MB in flies used calcium imaging, similar to previous techniques [8,9,14]. Noise can be intrinsic to the neuron (e.g., shot noise) or experimental (e.g., movement of the animal or sensor detection). Briefly, for each cell, we calculated its mean background response before odor onset, and selected a cell as being responsive if the mean for a set time after odor onset was above a threshold value determined by adding a certain number of background standard deviations (SDs) to the background mean. The number of background means, and thus the threshold value, is set by the *p*-value (*p*-value designates the significance threshold for deciding if a signal is true), which is 0.01 in the paper corresponding to 2.33 SDs. This is similar to *p*-values in our earlier work as well as work from other labs [100] (*p* = 0.05 above background mean) and [6] (*p* = 0.005 above air exposure). We outline the techniques and analysis we undertook to ensure that we detected the actual signal. Additionally, we also include analyses that outline how the results of the paper might depend on parameters such as the *p*-value or threshold for deciding if measured activity was a signal.

This section is broken into 3 subsections: mouse, flies, and dependence of the paper's results on measurement parameters.

**Mouse:** To determine if movement might contribute to reliable cells being classified as unreliable, we examined a video (uploaded to the repository and link provided in Data Availability section) of the subset of the cell population for the entire duration of the experiment, i.e., all odor exposures. In the video, we outlined the cells using the segmentation software that was originally used for identifying cells and getting their intensity. The outlines clearly show that the cells do not move out of the segmented outlines significantly, even when the brain moves. The plots in the accompanying figure (Fig 8B) show the total amount of movement in

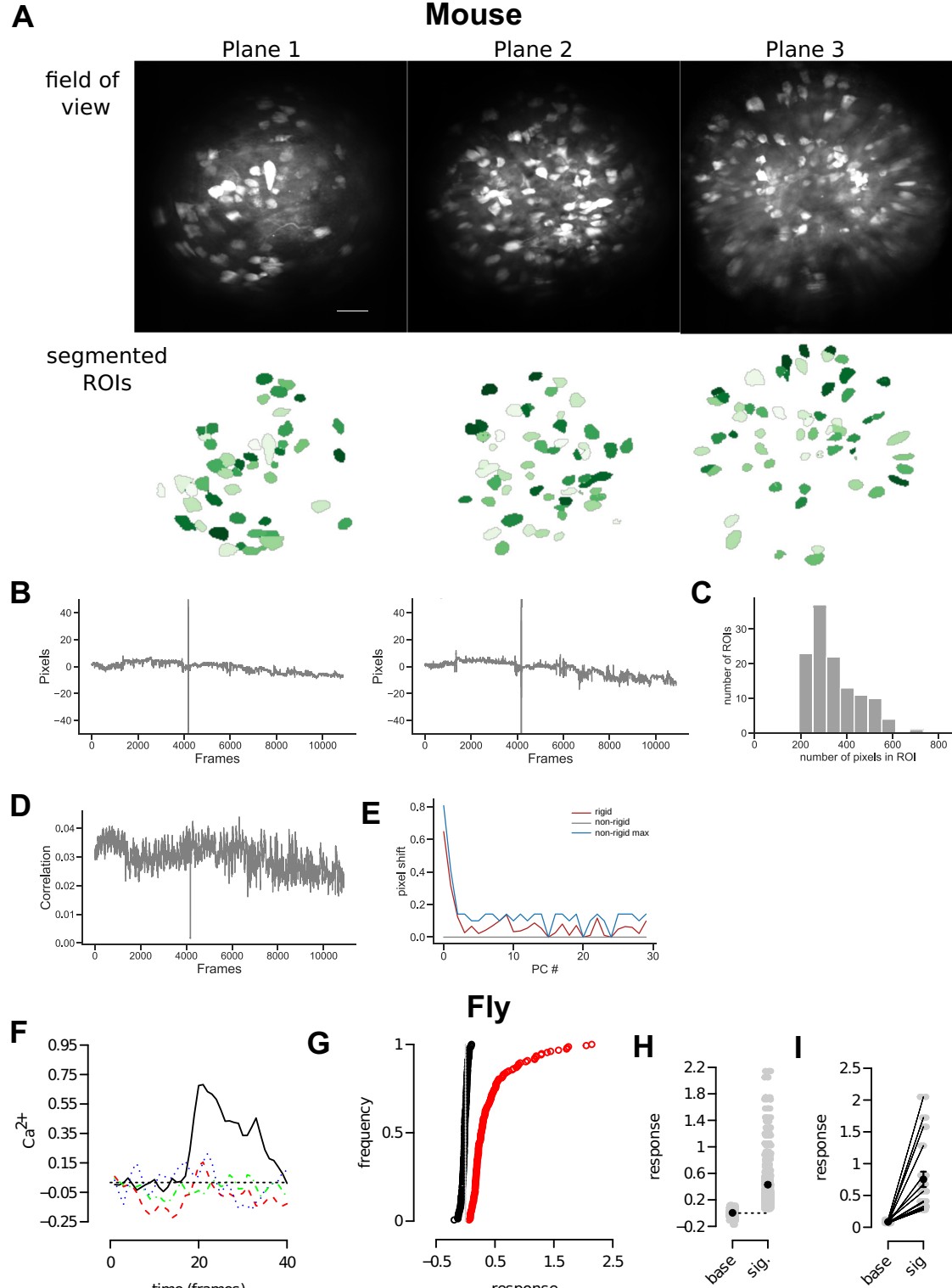

**Fig 8. Detecting signals with calcium imaging.** (A–E) Mouse calcium imaging data registration and segmentation. (A) Top: Maximum projection GRIN lens FOV of the T-series for each plane imaged. Bar: 100 μm. Bottom: Corresponding ROIs segmented. (B) Pixel shift in the x-axis (left) and y-axis (right) during registration at each represented by frames on the x-axis. (C) Size (in pixels) distribution of each of the segmented ROIs. (D) Peak of phase correlation between frame and reference image at each time point. (E) Magnitude of the shifts (both rigid and non-rigid) for each PC of the registered movie. (F–I) Fly calcium imaging analysis. (F) A few

example traces of different significant cells. The dotted line is $y = 0$ and indicates the value at which $Ca^{2+}$ activity is 0. (G) A cumulative distribution of the base (black) and signal (red) means. The signal follows a Gamma distribution as we showed in the paper (there is a little bit of a left skew), while the base mean is a very good fit for a Gaussian. The vertical dotted line ($x = 0$) denotes no response. (H, I) Plot of the base (left) and signal (right) means. There are 918 cells in all. The amount of overlap is about 90 cells for both responses, and 90 of the base means are above the lowest signal mean response. And, 90 of the signal means are below the highest base mean. We took the overlapping population of cells whose base means were above the lowest signal mean response, and re-plotted their and base and signals means while joining base and signal means for the same cells. We found that in these cells, all base means are lower than their signal means. Dotted line in (H) denotes the line ($x = 0$) denoting no response. See the Data Availability section for details on the 2 datasets used here: the fly main dataset (F–I) on Zenodo, and mouse dataset 164 (A–E) on Dandi. The data underlying the graphs shown in the figure can be found in S6 Data. FOV, field of view; PC, principal component; ROI, region of interest.

the x (left plot) and y (right plot) axis. This movement is negligible compared to the area of the cell. The distribution of cell sizes in Fig 8C shows that cell sizes range between 200 and 600 pixels, clearly an order of magnitude higher than the x and y movements. Thus, it is unlikely that unreliable cells emerge due to movement artifacts.

**Fly:** Although our analysis above captures the effect of motion artifacts, we want to say a bit more about the imaging analysis pipeline used for this data from [9,12]. In the pipeline, we identified in-plane cells by eye, clicked on them to place an ROI, and then a movie of the entire experiment was played underneath this ROI to visually confirm the cell in question did not drift out of plane for the entire course of the experiment. Indeed, there were many cells discarded because of movement/possibility of overlap with other cells etc. Typically, there were about 200 KCs in an FOV, and our yield of cells was 125 on average. Basically, we dissected a lot of flies to get a reasonable number of high-quality samples to analyze.

For flies, as the videos were not archived unfortunately, we tried an alternative method to ascertain if unreliable cells emerged due to experimental noise (Fig 8F–8I). Specifically, we calculated "noise" by looking at the fluctuations in dF/F in the pre-stimulus period. Signal variability here includes (i) photon noise; (ii) movement noise; and (iii) biological noise from spontaneous neural activity activity. So it is very likely an overestimate of biological variability.

We then compared the distribution of "baseline magnitudes" to the distribution of response magnitudes we measured during odor presentation (Fig 8G). We observed that less than 10% of response magnitudes were within the range of the baseline distribution (Fig 8G–8I). Moreover, of that 10%, none of the cells had their signals means lower than the base means. So experimental noise is a minor contributor to the results here.

Further, increasing the threshold for significance (Fig I in S1 Text), which would be analogous to silencing cells close to the background mean + SD (presumably the cells overlapping in the distribution), does not change the results of the paper. This is consistent with results from electrophysiological recordings of KC odor responses [8], which showed qualitatively similar levels of inter-trial variability.

**Analyzing the effect of *p*-value.** In Fig I in S1 Text, for both flies and mice, we show that either increasing or decreasing *p*-values, while changing the number of responsive cells, does not change the results of this paper. The supplement (analysis section) contains a detailed exploration of *p*-values and explanation of the analysis.

## Fig 1 and Fig A in S1 Text: Analysis of response characteristics

For both flies and mice, all trials contain randomly interleaved odors, e.g., an example run of the first 8 trials might have looked something like this: odor1, odor5, odor6, odor3, odor4, odor8, odor3, and odor1. Eventually, all odors are repeated an equal number of times. For each trial, $Ca^{2+}$ responses were recorded for the entire trial length that comprised 3 periods: the first period had no odor, a short odor exposure, and then a long period with no exposure.

The first period for mice lasted 10 s, and for flies also it was 10 s. Following that, in both cases, odor exposure lasted 1 s. The odor response was recorded as the mean responses for the 2.5 s following odor exposure (including the 1 s odor exposure period).

To determine if a cell's response was significant, we first calculated the mean and SD of the odor response values for the first period, i.e, the background period without any odor exposure. A cell was considered to have had a significant response if its odor response value was 2.33 SDs (background) above the background mean. Fig 1C shows the analysis of odor responses of 124 KCs across 42 trials of 7 odors. The *y*-axis measure of the percentage of cells represents the number of significant cells divided by total number of cells. The adjoining graph gives the averaged numbers for all trials for each odor. Fig 1E plots the reliability of each cell-odor pair versus the mean response size. The mean response size here is the expected value, i.e., $\sum_{trials}$ probability of response(trial$_i$)*response.size$_i$.

The calculations for the mouse PCx responses were similar. The mouse data in the main paper shows the responses of a set comprising 10 odors (2 controls) and 8 trials per odor, for 285 PCx cells. Here, too, odors were interleaved across trials. A response was considered significant if the mean response for the 10 time frames since odor onset was greater than 2.33 SDs above the background mean. Both SD and mean for the background were measured for the approximately 10 s before odor onset. The length of a time frame is 0.453 s.

For panels A and B in Fig 1 sup, we used a linearized form of the exponential equation, log($y$) = log($a$)+$bx$ to get the quality of the fit, while making sure that log errors were propagated properly.

## Fig B in S1 Text: Noise varies with signal size, supplemental item for Fig 1

The presence of trial-to-trial variability led us to wonder if the 2 subpopulations of cells could be differentiated not only on the basis of how frequently they respond but also on the variability of their signal. We had shown earlier that signal and frequency of response are linearly correlated. We next wondered if this relation might also extend to the noise or variability of signal across trials. Studies have used 2 measures of noise, coefficient of variation (CV) and the Fano factor (FF). While CV is defined as the SD over the mean, FF is variance (or SD$^2$) over the mean. For a Poisson distribution, wherein the variance is the same as the mean, FF would be 1.

Panel A in Fig B in S1 Text captures the relation of the signal to the noise through the CV and FF measures. Each gray dot represents the FF and CV for a single cell for 1 odor. The CV and FF measures for flies and mice were below 1. A significant difference between flies and mice was that in the fly, CV and FF measures were indistinguishable by cell frequency or a cell's reliability. In the mouse, on the other hand, cells with a lower trial frequency or reliability had a higher CV or FF measure. This finding reflects the fact that while in the fly, increases in average signal size kept track with a cell's SD (Fig B in S1 Text), in the mouse, although the 2 quantities were positively correlated, the rate of increase in SD in relation to signal size was less (Fig B in S1 Text). Thus, in flies, for most levels of reliability, there was no significant difference. In mice, however, cells with the highest reliability, had a lower level of CV and FF indicating that they might carry more robust information than other cells.

## Fig 2 and Fig E in S1 Text: Analysis of response, overlap, and reliability distributions

We use the cumulative frequency distribution of reliabilities as an illustration of how we made the cumulative plots. We first sorted all reliabilities in increasing order. We then assigned a value of one to the lowest reliability, and for each subsequent item we added 1. If several cells had the same reliability, say 3, we added the number of these cells to the counter. The counter

keeps track of the number of cells at or below this reliability value. Finally, we divided this counter by the total number of cells to get the cumulative frequency histogram values, and plotted them on a graph as shown in Fig 2B. The procedure we outlined is a nonparametric way of plotting frequency histograms, explained elegantly in [118]. We left out silent cells, i.e., cells that do not have a response on any of the trials.

### Fig 4, Fig C in S1 Text and Fig D in S1 Text: Analysis of correlations and overlap between odors

For the first part, to compare the similarity of odors in antenna and MB, we used Pearson's correlation coefficient. For the antenna, we calculated the correlation between all the odor pairs that were common to that dataset [68] (for OSNs) and [9] (for KCs). We then computed the correlations for the KC responses of these odor pairs. For the KC correlations of each odor-pair, we computed the correlation between 36 trial pairs (6 trials for each odor), discounting same trial odor-pairs, and took the mean of the 30 odor-pairs to be the correlation. This KC correlation was on the y-axis. Note, that the odor-pairs shown in Fig 4 are restricted to those whose correlations in the antenna and MB are within 0.2 of each other, since the Antenna responses are single trial responses and thus, might be more noisy than data collected over 6 trials in MB. Fig C in S1 Text shows that the qualitative results are the same even when all odor-pairs are considered. We also had access to 2 more datasets (with 5 more odors besides the main dataset), and we calculated correlations similarly for those datasets and have included them in this analysis (Fig 4A and 4B).

Fig 4D–4G gives the overlap between similar and dissimilar odor-pairs. For this analysis, we only used the main dataset. First, how did we determine similar and dissimilar odors? Fig E in S1 Text shows the distribution of odor similarities using correlation as a similarity measure. The correlation shown here is the correlation between the average response vectors, i.e., responses averaged across all trials. Odor-pairs that had a correlation less than 0.18 were considered dissimilar. In essence, it was 0.15 as there was no difference between 0.15 and 0.18. We chose 0.18 because when you choose vectors from a normal distribution and compare their correlations, 95% of them have a correlation $\leq 0.18$. Similarly, for the lower limit of similar odor pairs, we chose 0.35, so as to isolate a cluster of odors.

For every odor, we divided cells into their reliability classes, and calculated the cell's probability of response, which is the number of significant responses/total number of trials. Now, for the 2 odors A and B in an odor-pair, we can calculate the probability of the cell responding to A or B, and overlap is the probability that it responds to both, which is p(A).p(B). When we are considering odor-pair AB, the cell's reliability would be reliability of the cell for odor A, and vice versa for odor-pair B,A. The plots show the overlap calculated this way for every cell in each of the odor-pairs.

Fig C in S1 Text gives the overlap versus reliability measures for cells across all odor pairs, not just similar or dissimilar ones. In this figure, we can see that the average overlap over all odors, reduces the overlap of similar odors. This is because the overlap for a cell, when odors are dissimilar is low across all reliability classes (Fig 4G and 4H), and thus reduces the average overlap for cells with high reliability compared to the case of similar odors alone.

Panels D and E in Fig D in S1 Text gives the number of cells that are active from each reliability class for odors (calculated per odor and averaged across all sets of odors). For example, for reliability class or level of 1, for each odor set, we calculated the number of cells that were active for a reliability of 1, then averaged this number over all odors in the odor set. This procedure was repeated for every reliability level, for flies and mice.

Panels F and G in Fig D in S1 Text give the bias or specificity of cells within a reliability class or level. Bias or specificity is defined as the likelihood of a cell overlapping between 2 odors

normalized to (divided by) the overlap if all cells from within this class were chosen randomly. For the first part of calculating bias for the odor set, we took each cell in the odor set and calculated and assigned a 1 to the cell if it responded to odor A and odor B, and 0 otherwise. We then summed this quantity for all cells. For the second part of calculating bias with a randomly shuffled set, we computed the number of cells with reliability 1 for odor A and for odor B. With this number $n$, the probability of a cell being active for that odor is $n$/(total number of cells). Once this quantity is calculated, we can estimate the overlap. Note that we verified the calculation by also running simulations. The bias is then the (first quantity)/(second quantity) and gives the y-value seen in the plots. Also, note that the y-axis has a larger range for mice, as so does the x-axis. The bias trends are similar for both animals for the first 6 levels of reliability.

### Fig 5 and panels A and B in Fig H in S1 Text: Analysis of sparse coding

We compare 4 progressive groups of cells: the top 100, 75, 50, and 25 percentiles, to examine if these groups are more alike between similar odors. Here, when we say top percentile, we mean top percentile of responding cells. In flies, this would be roughly around 9 cells (since top 25% of 30 is around 8), and in mice, it would be around 28 cells.

Consider 2 similar odors, and that we would like to compare how similar the top 25% of cells for these 2 odors are. As a first step, we rank order the expected value of the odor response of each cell. We take the top 25% of cells for each odor, which could range from around 8 to 12 cells, because there will be some cells that respond to the both odors, while others do not. We then compute the cosine similarity of the average value of these cells for the 2 odors, and this value represents 1 gray point in the plot. Similarly, we compute the cosine similarity for all similar and dissimilar odor-pairs for each cohort of cells shown in Fig 5A and 5C. Here, we restricted dissimilar odors to include only those odors whose correlation (of average values) was ≤0.15. Similar odors had a correlation greater than 0.5. The number of dissimilar odor-pairs are fewer here compared to Fig 4, because in it we used an average of trial correlations, while here since we used average values for computing cosine similarities and percentiles, we also used average response values to compute correlations. The correlations of average response values are greater than the average of correlations across trials.

For Fig 5B and 5D, we took the top 25% and bottom 25% of responding cells for an odor, then we assigned them to the appropriate cell classes and counted that number. The points in gray are the numbers for the top 25% of cells, and the points in red are the numbers for the bottom 25% of cells for each odor. The averages are shown in black and darker red. And, we followed the same procedure in analyzing the fly and mouse cells, except for the mouse we only considered the top and bottom 10% of cells: since the absolute number of mouse PCs cells at 10%, was equal to the number of KCs at 25%.

For panels A and B in Fig H in S1 Text, we computed the cosine similarity among the 4 quartiles of size 25—top 100 to 75, 75 to 50, 50 to 25, and 25—for all odors. For both, flies and mice, the 25% quartile captures odor similarity information faithfully. When odor similarity is low, cosine similarity is low, and increases with increase in similarity of the population, indicating that they faithfully carry odor similarity information. The other 3 quartiles have flatter slopes indicating that they carry less information that can distinguish odors.

### Fig 4 and panel D in Fig C in S1 Text: Analysis of reliable, unreliable, and all cell responses with machine learning algorithms

To assess the ability of unreliable and reliable cells to classify odors, we isolated cells of each type, and then fed them to a decoder (based on simple machine learning algorithms). If either of the cell types were to carry odor distinguishing information, then the decoder should be

able to correctly classify odors when using these cells. We examined odor responses to 3 classes of cells: reliable, unreliable, and all significantly responding cells. To illustrate how we used the cells, we will use reliable cells as an example.

For each of the trials from an odor, we isolated all those cells that were reliable for that odor, and set the values of all other cells to 0. We repeated this procedure for every odor. Keep in mind that a reliable cell for one odor, might be unreliable for another, and thus would be positive for the first odor while being set to 0 for the second odor. There were some cells that were either silent or unreliable for all odors and did not play a part in classification. We then fed these updated odor responses to a simple decoder. For decoder classification, we split the data set into 2 groups: a training set and a test set. For all groups (reliable, unreliable, and all cells), the training set contained 80% of trials, and the test set contained 20% of trials. Finally, to ascertain that the results did not arise because of random effects on how the test and training sets were chosen, we used $k$-fold cross validation [119], in which we set fold size to 20%. At the end, the presented results were averaged.

Classifier performance was reported using 2 measures: accuracy, and multiple class area under the receiver operator curve (AUC) according to [120]. Both measures were calculated on the test set. Accuracy is the fraction of trials in which the predicted odor and the actual odor are the same. The second measure, which we call all-to-all (A2A) AUC, is computed for all pairs of odors, and the numbers plotted in Fig 4 are the average AUC for all odor pairs. The individual pairwise odor AUC was calculated as the average of $A(i|j)$ and $A(j|i)$, where $A(i|j)$ is the probability that a random trial of odor $j$ has a lower probability of being classified as odor $i$ than a random trial of odor $i$. A high score (maximum 1) denotes that the 2 odors $i$ and $j$ are easily separable. A low score indicates a high likelihood of them being mistaken for each other.

For AUC calculation, the test set was used for obtaining a matrix of odor-predictions *Pred. mat*. The rows were the presented odors, and the columns were the set of odor classes. Each entry is the probability of the odor $i$ being classified as odor $j$. The AUC score for odor $i$ and $j$ as mentioned earlier is $AUC(i|j)$ and is based on the following equation:

$$AUC(i|j) = \frac{S_i - n_i(n_i + 1)/2}{n_i n_j} \qquad (4)$$

The entries in this equation will become clearer if we explain the algorithm used. We take all the test odors (trial responses in the test set) from the odor classes $i$ and $j$ that were presented, and calculate vectors $f_i$ and $f_j$. $f_i$ contains the entries (from *Pred.mat*) for all the test odors from class $i$ and their probability of being classified as odor $i$. Similarly, $f_j$ contains the test odors from class $j$ and their probability of being classified as odor $i$. $n_i$ and $n_j$ are the number of entries in $f_i$ and $f_j$. We rank order the entries of $f_i$ and $f_j$, and then $S_i$ is sum of all the ranks for the entries from $f_i$. We repeated this procedure for $AUC(j|i)$) and took the average so $A(i, j) = (AUC(i|j) + AUC(j|i))/2$. The AUC score for the dataset is the average of all odor pairs $(i,j)$.

We used 3 types of machine learning algorithms (2 with a linear kernel and 1 nonparametric): LDA, kNN, and SVM or support vector machines with a linear kernel [119]. The scores and the training:test set ratios are given for each of the decoders in Table C in S1 Text.

As a test case for the A2A AUC, we also computed AUC values for odor-pairs that were either similar or dissimilar (panel D in Fig C in S1 Text). We found that similar odors had a low average AUC while dissimilar odors had a high average AUC. The AUCs depicted in panel D in Fig C in S1 Text were calculated for all cells. For this calculation, the classifier that we employed was kNN on the mouse dataset collected in this paper. We observed similar results with the linear SVM classifier and the fly dataset.

## Modeling and theory

Fig 3 describes the model used for understanding the mechanism of generating a stochastic code. Fig 6 describes the learning and discrimination model that explains how the stochastic code (using data in Fig 1) helps in discriminating similar odors.

### Fig 3: WTA mechanism responsible for a stochastic code

We used a linear rate firing model of the olfactory circuit from the antennal lobe to MB, similar to previous models of the olfactory circuit [70,121]. For PN activity and connections of PNs with KCs, we used connection characteristics revealed by an analysis of previous studies by [41] and listed in the paper's supplement. In the antennal lobe, we mimic odor activity of PNs by sampling from an exponential distribution, based on analysis of [47]. For the connectivity matrix from PNs to KCs, we used the same parameters as data distributions from [59]. We generated the connection matrix in the following way: For each KC, we generated the number of claws it has by sampling from a Binomial distribution with $p = 0.85$, and $n = 8$. Then, each of these claws would get connections from one of 50 PN types, and the PN -type was chosen by sampling from a hypergeometric distribution with parameters $n = 50$ and $a = 0.07$ (again, based on [41,59]). Finally, for the strength of the synapses, we sampled from a Gamma distribution with shape and scale factors of 4 and 4 based on [60].

As mentioned in the main text, we examined 2 types of models (a single synapse and multi-synapse model) to test how the fly circuit could generate the stochastic code that helps with discriminating similar odors. The 2 models differ in their connectivity within the WTA circuit and helped us pinpoint the role of connectivity in generating the stochastic code. In the single synapse model, APL made a single synapse with each KC, where the strength of each synapse was equivalent to the synapse strength as per [55,61]. Or, equivalently, APL made multiple synapses with every KC, but, all synapses had the same noise (correlated noise). In the multi-synapse model, APL made multiple synapses with every KC, and the synapses were independent of each other. So, noise in these synapses was uncorrelated. The equations describing both models is thus essentially similar, except for the equation detailing noise in APL→KC connections.

For the next step of the WTA involving KCs and APL, we used data from [55,61], who showed that in both directions, KCs ↔ APL, the number of contacts are correlated. We varied the number of APL→KC feedback synapses in the range of 5–38 per KC, with a mean of about 15 to 20 synapses following [55].

A parameter not listed in these studies, but, nevertheless important for the model is the APL→KC inhibition gain, which determines the number of top KCs that remain active after the WTA mechanism is active. As we show in our parameter exploration and further analysis (panel A in Fig K in S1 Text), we examined a range of APL gain that gave the top % of active cells as being in the range of 6–11 KCs, when there was no noise in the system.

**Model:** The model can be described by the equations that capture the various relationships. Each KC, gets input from a collection of PNs. Therefore every KC, $K_i$ is the sum of its inputs from all projection neurons $P_j$'s.

$$K_i = \sum_j P_j * s_{ji} \tag{5}$$

where $s_{ji}$ is the strength of the synapse from $P_j$ to $K_i$. For most PNs, the synapse strength $s_{ji}$ is 0, except for a select few (around 6). Activity at KCs can also be thresholded, and we use a

simple threshold $t$, wherein,

$$K_i = 0 \text{ if } K_i < t \text{ else } K_i = K_i - t \tag{6}$$

In the multi-synapse model to be described below, we found that a threshold that suppressed the bottom 10% to 20% of cells had no effect on our results. For reducing computation time, we did not include a threshold for parameter explorations to reduce computational time, but it is included for the examples (Fig K in S1 Text).

All KCs send excitatory connections to the APL neuron. APL neuron activity is given by

$$APL = \sum_i K_i * (s_{K_i A}), \tag{7}$$

where each $s_{K_i A}$ is the sum of several synapses since each KC makes several ($k$) synapses with the APL neuron. So,

$$s_{K_i A} = \sum_k s_{K_i A_k} \tag{8}$$

APL, in turn, feeds back and suppresses KC neurons, and this activity can be captured as the APL inhibition of KCs.

$$APL_{inhibition} = APL * s_{AK_i} * APL_{gain} \tag{9}$$

$$\text{Therefore, } K_i = \sum_j P_j * s_{ji} - APL_{inhibition} \tag{10}$$

$$K_i = \sum_j P_j * s_{ji} - APL * s_{AKi} * APL_{gain,} \text{ where } s_{AK_i} = \sum_l s_{AK_{il}} \tag{11}$$

Here, each $K_i$ receives $l$ synapses from APL. $l$ and $k$ are correlated.

**Noise**: To examine the effects of noise on the circuit, we introduced noise within various components, similar to previous investigations of the olfactory circuit [70,121]. For each component, we added multiplicative noise sampled from a normal distribution, e.g., if a PNs firing rate was $P_j$, the noisy version of the PN, $P_{noise}$ was $P_{noise_j} = P_j(1 + \eta(0, \sigma))$, where $\sigma$ designates the amount of noise. If the noise level was 10%, $\sigma = 0.1$. We explored noise in the following basic (only in 1 component) scenarios: (i) Noise in the PNs; (ii) Noise in PN→KCs; (iii) Noise in KC → APL connections, (iv) Noise in APL → KC connections, (v) Noise in APL, and (vi) Noise in KCs.

To give an illustration of how this would affect the model, we demonstrate how the equations of PN noise would change KC activity.

$$P_j = P_j \{1 + \eta(0, \sigma)\} \tag{12}$$

$$\text{which leads to } K_i = \sum_j P_j * \{1 + \eta(0, \sigma)\} * s_{ji} \tag{13}$$

$$K_i = \sum_j (P_j * s_{ji} + P_j * \eta(0, \sigma) * s_{ji} \tag{14}$$

$$K_i = \sum_j P_j * s_{ji} + \sum_j P_j * \eta(0, \sigma) * s_{ji} \tag{15}$$

The first part of Eq 15 is the same as Eq 5. The second part is similar to the first except it is multiplied by $\eta(0,\sigma)$. When sampled multiple times, this quantity will tend to 0, since the expected value is 0 and the more samples we take the closer the average will be to 0.

Note that the single- and multi-synapse models differ in how they treat noise at APL↔KC synapses. In the single synapse model, noise in all the synapses is correlated, while in the multi-synapse model, it is correlated. The difference lies in the $APL*s_{AK_i}$ part of Eq 11. In the single-synapse model, it is $APL*\{1 + \eta(0,\sigma)\}\sum_l s_{AK_{il}}$ while in the multi-synapse model, it is $APL*\sum_l s_{AK_{il}}\{1 + \eta(0,\sigma)\}$.

**Parameter exploration:** The parameters used for the different models in Fig 3 are as follows. For all the models, the number of PN types was 50, and the number of KCs was 150. The number of odors was 6, with 6 trials per odor. The APL inhibitory feedback gain was set so that only the top 6% to 11% of cells (we explored this range in increments of 1) were active without noise present.

We explored parameter space to isolate the components that when subject to plausible noise, produce observed behavior. The second purpose of parameter exploration was to explore parts of the model that have not been experimentally measured yet (such as the connectivity details of the APL feedback to KCs or the amount of noise in a component and its effect on function) and test the parameter regimes under which we observe the stochastic code and other results.

We provide a detailed explanation of the various parameters used and their effect on the models in the supplement (modeling and theory) section and accompanying figures: Fig J in S1 Text, Fig K in S1 Text.

## Fig 6: A model of learning and discrimination based on the fly olfactory learning network

The objective of this model was to examine the contributions of reliable and unreliable cells towards discrimination. To have a realistic estimate of how the cells would work in an actual system, our model closely mimicked the fly association network detailed in the main text accompanying Fig 6, and depicted in Fig 7. For more in-depth detail, please refer to [31].

There are 2 elements to the model. The first is the contribution per cell, and the second is the learning in the network, which is done by plasticity at the synapses between KCs and MBONs. We approximate the fly MBONs to 2 types: approach and avoid; the real fly has about 15 MBON types falling into either the approach or avoid categories. Learning is implemented in 2 stages that reflect the amount of training: normal, which in the fly case would be around 12 trials, and extended, which would be about 100 trials. With normal training, we consider the discriminatory power of the cell using Eq 1. Initially, we assign equal weights to the connections from every KC to both MBONs. These weights undergo a change depending on whether they are from reliable or unreliable cells. The change in weights is governed by Eq 3. Since reliable cells respond consistently and with a high response value their weight approaches 0 very quickly, i.e., by the end of the normal training period. Unreliable cell synapses, for the same reason, undergo little change in their synaptic weights. The overall discrimination—which we call useful discrimination—as shown in Eq 1 is the sum of all cells. It is $D(A, B) = D(A,B)_R + D(A,B)_{UR}$. Since reliable cell synapses (from KCs to either the approach or avoid MBON depending on whether it is punishment or reward) are close to 0, $D(A,B)_R$ is high, and as unreliable cell synapses are unaffected, $D(A,B)_{UR}$ is low. The contribution of $D(A,B)_{UR}$ is significant, however, since there are so many unreliable cells.

When learning is extended over 100 trials, there is no significant change in the synapses of reliable cells (they are already 0), but those of unreliable cells undergo a large change and are

close to 0. Therefore, the contribution of unreliable cells to discrimination is also high now. One might think that since Eq 1 depends on the probability of response, it would be low. But, as a population, unreliable cell contributions are high as there are many more unreliable cells than reliable (29 compared to 5 for flies and 43 compared to 4 in mice). So, even if the unreliable cell responds in only 1 or 2 trials, there are at least 5 times as many cells, and their contribution is substantial.

For Fig 6, for easy calculation, we set the weights of all reliable cell synapses to 0 after normal training, and the weights of all unreliable cell synapses to 0 after extended training. Unreliable cell synapses remained nearly unaffected by normal training.

For Fig 6E and 6F, we first isolated all cells with a specific reliability for an odor, e.g., reliability class 1 would be all cells with reliability 1. Second, we silenced all other cells, and carried out an analysis similar to panels A–D and calculated useful discrimination for normal and extended training, and measured useful discrimination increase as the difference of extended versus normal training. This procedure was followed for flies and mice.

Finally, one of the characteristics that we tested for determining the plausability of the WTA model was to test if it produced the results that we observed with our learning model. These results and the parameters explored are presented in the supplement and Fig K in S1 Text (the figure shows examples of parameter sets that were "successful,", i.e., those that generated outputs similar to the results observed in Figs 1, 2 and 6).

## Supporting information

**S1 Text. Fig A. Odor response sizes in the fly MB and mouse PCx cells increases with reliability. Fig B. Responses in Mouse PCx cells are more variable compared to flies. Fig C. Reliable cells preserve odor-similarity better than unreliable cells. Fig D. Unreliable cells are more likely to respond differently between similar odors compared to reliable cells. Fig E. The probability of overlap cell increases from similar to dissimilar odors. Fig F. Unreliable cells are a composition of cells with different levels of reliabilities. Fig G. Noise in the winner-take-all mechanism produces a stochastic code. Fig H. Sparse coding does not improve discrimination ability for similar odors. Fig I. The effect of the significance levels on discrimination analysis. Fig J. The effect of the significance levels on discrimination analysis. Fig K. Fly circuit model parameter explorations. Table A. Possible contributions by cell *x* towards discrimination. Table B. The top 40 parameter combinations (out of 44,217) of the single synapse model that produced results most similar to the ones observed in Fig 1C and 1D. Table C. Results of the performance of 3 linear classifiers/ decoders on the fly and mouse datasets. Table D. The response characteristics (Fig 1) for all the flies that were examined. Table E. The response characteristics (Fig 1) for all the mice that were examined.**
(PDF)

**S1 Data. Data that underlies Fig 1.**
(XLSX)

**S2 Data. Data that underlies Fig 2.**
(XLSX)

**S3 Data. Data that underlies Fig 4.**
(XLSX)

**S4 Data. Data that underlies Fig 5.**
(XLSX)

**S5 Data. Data that underlies Fig 6.**
(XLSX)

**S6 Data. Data that underlies Fig 8.**
(XLSX)

**S7 Data. Data that underlies Fig A in S1 Text.**
(XLSX)

**S8 Data. Data that underlies Fig B in S1 Text.**
(XLSX)

**S9 Data. Data that underlies Fig C in S1 Text.**
(XLSX)

**S10 Data. Data that underlies Fig D in S1 Text.**
(XLSX)

**S11 Data. Data that underlies Fig E in S1 Text.**
(XLSX)

**S12 Data. Data that underlies Fig F in S1 Text.**
(XLSX)

**S13 Data. Data that underlies Fig H in S1 Text.**
(XLSX)

**S14 Data. Data that underlies Fig I in S1 Text.**
(XLSX)

## Acknowledgments

We thank the reviewers for helpful suggestions and constructive criticism. We appreciate Chuck Stevens' feedback and support for the project. We thank Andrew Lin, Rob Campbell, and Kyle Honneger for their quick responses and for generously sharing data from their previous publications. We also thank Terry Sejnowksi and Jorge Aldana for the computational resources needed for carrying out the data analysis.

## Author Contributions

**Conceptualization:** Shyam Srinivasan, Saket Navlakha.

**Data curation:** Shyam Srinivasan, Simon Daste, Glenn C. Turner, Alexander Fleischmann.

**Formal analysis:** Shyam Srinivasan, Saket Navlakha.

**Funding acquisition:** Glenn C. Turner, Alexander Fleischmann, Saket Navlakha.

**Investigation:** Shyam Srinivasan, Simon Daste, Glenn C. Turner, Alexander Fleischmann, Saket Navlakha.

**Methodology:** Shyam Srinivasan, Glenn C. Turner, Alexander Fleischmann, Saket Navlakha.

**Project administration:** Shyam Srinivasan.

**Resources:** Glenn C. Turner, Alexander Fleischmann, Saket Navlakha.

**Software:** Shyam Srinivasan.

**Supervision:** Shyam Srinivasan, Glenn C. Turner, Alexander Fleischmann, Saket Navlakha.

**Validation:** Shyam Srinivasan, Simon Daste.

**Visualization:** Shyam Srinivasan, Saket Navlakha.

**Writing – original draft:** Shyam Srinivasan, Saket Navlakha.

**Writing – review & editing:** Shyam Srinivasan, Simon Daste, Mehrab N. Modi, Glenn C. Turner, Alexander Fleischmann, Saket Navlakha.

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
