## [Editor Report · Decision Letter 0]

28 Mar 2022

Dear Dr Srinivasan, 

Thank you for submitting your manuscript entitled "Stochastic coding and implications for odor discrimination" for consideration as a Research Article by PLOS Biology.

Your manuscript has now been evaluated by the PLOS Biology editorial staff as well as by an academic editor with relevant expertise and I am writing to let you know that we would like to send your submission out for external peer review.

Once your full submission is complete, your paper will undergo a series of checks in preparation for peer review. Once your manuscript has passed the checks it will be sent out for review. To provide the metadata for your submission, please Login to Editorial Manager (https://www.editorialmanager.com/pbiology) within two working days, i.e. by Mar 30 2022 11:59PM.

If your manuscript has been previously reviewed at another journal, PLOS Biology is willing to work with those reviews in order to avoid re-starting the process. Submission of the previous reviews is entirely optional and our ability to use them effectively will depend on the willingness of the previous journal to confirm the content of the reports and share the reviewer identities. Please note that we reserve the right to invite additional reviewers if we consider that additional/independent reviewers are needed, although we aim to avoid this as far as possible. In our experience, working with previous reviews does save time. 

If you would like to send previous reviewer reports to us, please email me at ialvarez-garcia@plos.org to let me know, including the name of the previous journal and the manuscript ID the study was given, as well as attaching a point-by-point response to reviewers that details how you have or plan to address the reviewers' concerns. 

Given the disruptions resulting from the ongoing COVID-19 pandemic, please expect some delays in the editorial process. We apologise in advance for any inconvenience caused and will do our best to minimize impact as far as possible.

Kind regards,

Ines

--

Ines Alvarez-Garcia, PhD

Senior Editor

PLOS Biology

---

## [Decision Letter · Decision Letter 1]

23 May 2022

Dear Dr Srinivasan,

Thank you for your patience while your manuscript entitled "Stochastic coding and implications for odor discrimination" was peer-reviewed at PLOS Biology and please accept my apologies for the delay in providing you with our decision. Your manuscript has been evaluated by the PLOS Biology editors, an Academic Editor with relevant expertise, and by two independent reviewers.

The reviews are attached below. As you will see, the reviewers find your conclusions novel and potentially interesting, however they also raise several conceptual, experimental and structural concerns that would need to be thoroughly addressed in order for us to consider the manuscript for publication. After discussing the comments with the Academic Editor, we would like to invite you to submit a revision that addresses all the points raised by the reviewers.

Given the extent of revision that would be needed, we cannot make a decision about publication until we have seen the revised manuscript and your response to the reviewers' comments. Your revised manuscript would need to be seen by the reviewers again, but please note that we would not engage them unless their main concerns have been addressed. 

We appreciate that these requests represent a great deal of extra work, and we are willing to relax our standard revision time to allow you 6 months to revise your study. Please email us (plosbiology@plos.org) if you have any questions or concerns, or envision needing a (short) extension.

**IMPORTANT - SUBMITTING YOUR REVISION**

3. Resubmission Checklist

a) *PLOS Data Policy*

b) *Published Peer Review*

If your manuscript contains blots, please provide the original, uncropped and minimally adjusted images supporting all blot and gel results reported in an article's figures or Supporting Information files. We will require these files before a manuscript can be accepted so please prepare them now, if you have not already uploaded them. Please carefully read our guidelines for how to prepare and upload this data: https://journals.plos.org/plosbiology/s/figures#loc-blot-and-gel-reporting-requirements

Sincerely,

Ines

--

Ines Alvarez-Garcia. PhD

Senior Editor

PLOS Biology

Reviewers' comments

Rev. 1:

In this manuscript, Srinivasan et al tackle an interesting question: is there a role for the many cells that respond weakly and unreliably to stimuli? They examine responses of 3rd order neurons in the fly and mouse olfactory system to different stimuli, and propose that unreliable neurons have a role in differentially representing similar odors. There are some interesting ideas and analysis in this manuscript, but I think several issues should be addressed before strong inferences can be made about the results.

Conceptual issues:

1. I find the overall framing of the manuscript to be somewhat unclear. The initial paragraphs of the introduction set up the paper as a test of the hypothesis that a sparse-coding-like scheme is implemented in the olfactory system, but the remainder of the paper lacks such a clear direction. At some points the reader is led to believe that this is a paper about possible computational roles for trial-to-trial variability, while at others the focus is again on a comparison to sparse coding. The current organization of the figures likely contributes to the lack of clarity; moving the discussion of different strategies for fine discrimination, and what is currently Figure 7, to the Introduction might help guide the reader.

2. Some discussion of the idea that the brain might perform sampling-based probabilistic inference (eg. work of Lengyel, Pouget) would be useful, particularly in how this idea might relate to the coding scheme proposed here. On first reading, these ideas appear compatible, and both ascribe a computational role to neural variability.

3. The discussion of the ubiquity of trial-to-trial variability in neural systems---as well as its possible benefits---is a bit vague; it should be revised to specifically address the question of variability in early sensory coding. In doing so, it is important that the authors distinguish between the different timescales over which responses may vary, and the circuit mechanisms to which variability may be attributed. In that vein, it might be useful to cite a review on longer-timescale representational drift, such as Rule et al 2019 or Masset et al 2022. It is also not sufficiently clear whether the phenomenon of stochastic resonance is directly relevant.

4. The current title could lead the reader to believe that this is the first paper to propose a concrete computational role for noise in neural systems. This is clearly not the case, and the authors might consider adopting a more specific, precise title.

5. The discussion around Figure 6 seems needlessly confusing. It would be more useful to describe the model and show the result in Figure 6 without the example numbers or Table 1.

6. The conceptual discussion of models for fine discrimination in Figure 7 seems overly simplistic, as it treats the code as roughly binary and does not address the structure of noise. In particular, this is important since noise scaling or addition could blur any of the distinctions among representations of different odors.

Methodological issues

1. It is vital that the authors provide more justification for the criteria used to classify cells as "significantly responsive" and "reliable." Indeed, the authors themselves discuss the fact this distinction is "somewhat arbitrary." Whether these choices are merely useful for exposition or profoundly affect the qualitative results should be made clearer. One possibility would be to show how sensitive subsequent analyses are to the thresholds chosen (to define responsive vs not responsive). For a large part of the paper, the definition of reliability seems to be based on binary "respond" or "not" - rather than differences in firing rates.

Related: the view of defining unreliable cells in a SINGLE trial is a bit confusing because you can only know whether a cell is "unreliable" or not from multiple trials.

2. The motivation for using multiple classifier types in the decoding analysis is unclear. Why not just use a linear decoder, for which one would have a clear idea of what features of the response are relevant? Also, the method used to isolate reliable and unreliable cells for the discrimination analysis seems peculiar. Instead of setting the responses of cells in some sub-population to zero and then performing the decoding analysis, why not just exclude that sub-population? Also, why is the cross-validated accuracy not used?

3. Figure 4C. This seems like a really bad comparison: if you look at the y axis, the range of values for similarities is very small and around 0. This is likely simply a consequence of unreliable and weak responses - so saying there is a lack of correlation is not justified. It would be more interesting and significant if the range in the y axis had been large and then there is no correlation.

4. The statement that "a winner-take-all mechanism is required for generating stochastic codes" is too strong; the results here merely show that such a mechanism is sufficient within the given model, while a subset of others is not. This is necessarily a model- and search space- dependent statement. As far as I can tell, the role of the inhibitory feedback synapse reliability is experimented only numerically - no analytical or conceptual reason given for why that synapse is the one. Also, could it be due to the assumptions of particular connectivity? For example, how will this WTA mechanism work when the feedback signal comes from a population of inhibitory neurons, rather than a single (or a few) APL-like cells? Does stochasticity across different inhibitory neurons need to be coordinated?

Miscellaneous points:

1. The quality and clarity of the figures could be improved, as they are often difficult to understand without careful reading of the caption.

2. Results of Figure 1E&H are entirely expected, right? That is, only cells that are weakly responding are likely to be unreliable (a simple consequence of peri-threshold variability?)

3. Figure 2: why is overlap calculated this way, rather than at a population level? something to do with number of simultaneously-recorded neurons? That brings up the question: how many neurons were simultaneously recorded, and are trial identities the same for all neurons (ie, simultaneous vs pseudo timed)?

4. Figure 2B/E: how will these curves change when the number of trials used to calculate reliability increases?

5. Figure 3AB: I find the fully linear model to be strange if you don't include a threshold nonlinearity, which is the essence of neural function.

6. The color bar in Figure 3H seems highly skewed, it looks nearly uniform.

7. In the discussion of trial-to-trial variability as a driver of exploration, it would be useful to cite a more recent review on song learning, e.g., Chen & Goldberg, Current Opinion in Neurobiology, 2020. This is relevant only if the authors retain some discussion of the role of variability in behavioral learning.

8. On page 7, the authors incorrectly characterize the Gamma distribution as the maximum entropy distribution with constrained mean E(X) and log-mean log E(X). The correct statement is that the mean E(X) and mean-log E log(X) are constrained.

9. Figure 5E: I find the results counterintuitive - the sum of everything should always do better than a subset!

10. Page 7, around line 25 or so. "This measure is also the reliability measure of the cell used in previous figures (Figs. 1,2)." why add a different name for the same measure?

Rev. 2:

This paper analyses the stochasticity of olfactory responses in the fly mushroom body and mouse piriform cortex, and makes the striking and counterintuitive claim that unreliable, noisy responses can actually be functionally beneficial in learned sensory discrimination. When two similar sensory stimuli evoke activity in overlapping populations of neurons, the most reliably-responding cells are the most likely to overlap between the two stimuli, which makes them useless for discriminating between the stimuli. In contrast, unreliable cells are less likely to be active for both stimuli and thus can serve as a substrate for discriminating the two stimuli. This result provides an exciting new perspective on sensory coding for learned discrimination and would be well worth publishing after some revisions.

Particular strengths of this work:

- It integrates analysis of experimental data from both fly and mouse

- It draws insights from these data about how the stochasticity and statistical distributions of the responses might be computationally useful

- It's creative and proposes an original idea

The principal areas that need to be strengthened before publication are:

- Show that unreliable cells actually contribute to model performance at discriminating similar odors

- Establish how much of the observed stochasticity in responses is real neuronal variability vs. arising from experimental noise

- Clarify if APL->KC noise is really the only way the observed stochasticity could arise

1. The most important conclusion is that unreliable cells allow similar odors to be distinguished because their responses are uncorrelated even for similar odors. However, their only measure here (Fig 6) is discriminability - D(A,B) - which isn't a measure of accurate discrimination per se but rather how different are the output weights of a certain KC toward the avoid vs. approach output neurons. It's not clear if this discriminability measure translates into actual discrimination. Maybe the unreliable cells are too few in number for their superior discriminability to improve the actual discrimination?

To justify their conclusions, the authors need to extend this result to show that a model circuit performs better at learned odor discrimination with unreliable cells than without, and that it performs better when you delete reliable cells, than when you delete unreliable cells. According to the authors' predictions, such a result would only be true for similar odors and only with extended training. This would be like Fig 5E,F, except separating out similar/dissimilar odors, separating out normal/extended training, and using a biological learning algorithm instead of decoder models.

2. About the unreliability of responses: Some of this could come not from neuronal variability, but from signal detection failures or detection noise in the calcium imaging, or the brain moving between trials. Some points to consider:

- Campbell et al 2013 used GCaMP3 which is much less sensitive than more recent indicators. So the percentages in Fig 1 are probably an underestimate of the true responsiveness.

- Presumably there is some threshold for declaring that a cell is responsive or not - but sometimes a cell might genuinely respond but just fall below the analysis threshold

- The brain probably moves between trials. They have applied non-rigid motion correction but this probably isn't perfect (and it wouldn't fix z-motion). How sure are the authors that an "unreliable" cell didn't just slightly move outside the ROI during a few of the trials?

Presumably, the authors believe that experimental artifacts can't explain all of the inter-trial variability. Can they give some justification for an upper bound on how much of the inter-trial variability is experimental noise rather than true neuronal variability (e.g., 5%, 20%, 50%, 100% etc)? This could be based on some reasonable guesstimates of how much photon noise there was, how bad the motion artefacts were, etc. This quantification is important because if the true responses are less variable than the observed responses, then perhaps noise in PNs+PN->KC synapses (Fig 3G) is enough to produce the true variability (no need to invoke noise in APL->KC synapses). It's also important because the stochasticity is the principal experimental result in the paper.

3. About the role played by noise in APL-KC synapses (Fig 3): It seems the principal role is by making the effective spiking threshold of a KC vary from trial to trial - sometimes the excitation from a particular odor is enough to make the KC spike, sometimes not. Couldn't this arise not (only) from noise in APL->KC synapses, but rather (or also) from noise in the integration of synaptic excitation? - see Groschner et al 2018 Cell - the subthreshold depolarisation and then spiking (or not) of KCs certainly seems very noisy and variable across trials. (Of course, the measurements in Groschner include noise from both excitation and inhibition, but the point is we can't tell which one it is.) I couldn't tell from the Methods, but I suspect that in the rate coding model, inhibition and thresholding both take the same form (subtraction) - or perhaps are even literally the same thing since I don't see mention of a separate thresholding step. (Note that noise in KC integration would be a distinct source of noise from noise in PN->KC synaptic strength, which the authors argue is not sufficient to produce observed unreliability. The difference seems to be that noise in PN->KC synaptic strength gets averaged out across the many PN inputs for each KC, but noise in the KC's intrinsic properties doesn't.) I don't think the authors can conclude that only APL-KC noise (rather than noise in KC synaptic integration in general) can explain the observed stochasticity.

4. Another major conclusion is that unreliable cells have less correlated responses to odors (Fig 4). I'm having trouble evaluating the conceptual novelty of this conclusion. Could it simply be mechanically true that when any cells respond unreliably, they're less likely to overlap for different odors, almost by definition? If every cell responds completely randomly, would you still expect that cells that respond more often would have more overlap than cells than respond less often? Similarly, for Fig 4C, could it simply be that unreliable cells are responding rather randomly, thus the population responses of unreliable cells are uncorrelated because the responses are pretty much random? But perhaps this is precisely the authors' point? They should clarify whether they interpret this result as "unreliable cells must inevitably be less overlapping, by definition" or "it's not necessarily true that unreliable cells are less overlapping, but they are in this particular dataset, i.e. it's a feature of fly/mouse coding schemes that unreliable cells are less overlapping". They could perhaps demonstrate which of these is the correct interpretation by creating fictitious random responses not based on real olfactory inputs, and showing that the unreliable cells are/aren't less overlapping.

5. Fig 4-6: the authors split cells into reliable and unreliable classes. But as far as I understood from Fig 1 and p.36, reliability is only defined relative to a specific odor. What do they do for cells that are reliable to odor A but unreliable to odor B? Are these excluded from analysis? Also, it's not clear to me for Fig 4B,C that it's valid to draw a scatter plot and trend line when every dot represents a different set of KCs (since they represent different odor pairs that must have a different set of reliable and unreliable KCs) - the authors need to be more convincing of the rationale for this analysis.

6. Related to this, for Fig 4D-G: How is overlap affected by whether a neuron has the same vs. different reliabilities for A vs. B? (Is this a straightforward result of the definition of overlap =P(A)*P(B) as per my point #4 above or is it more complicated?)

Fig 4D-G - Shouldn't overlap probability be the y-axis, and reliability the x-axis? It seems like their argument is that overlap depends on reliability, not that reliability depends on overlap. The way they describe "slope steepness" in the text also seems to require swapping the x and y axes.

7. I don't understand the conclusions drawn from Fig 5A,C and Fig S8.

- "The top 25% of cells are more similar than the rest of the population (i.e., they provide no extra discriminatory information)." - I don't see how the parenthetical follows from the first part of the sentence

- Fig S8 title "Sparse coding does not improve discrimination ability for similar odors." I don't see how this follows from the figure

- "we found that the cosine similarity of odors did not decrease from the top 50% of responsive cells to the top 25% of cells for similar odors (top circles), which would be expected if sparse coding improves discrimination." I don't think the first part of the sentence is a common prediction of the idea that sparse coding improves discrimination - the latter idea is usually that if few cells are active in general, then two different odors would likely have relatively few overlapping cells - it's not that the strongest-responding cells are supposed to form the most distinctive patterns. Moreover, cosine similarity doesn't necessarily correspond to the accuracy of discrimination (this comes back to my point #1).

8. "As the reliability of individual KCs increased, their mean response levels increased exponentially" - I'm not sure this is the best way to describe it. In both mice and flies, the last point (100% reliability) is clearly an outlier and one would also think a priori that it would be an outlier. The first few points actually seem to increase linearly - which is actually what you'd expect purely mechanically if the cells have the same response when they do response, but when this is averaged with the non responses, the mean response across trials is lower for the more unreliable cells. If there's a break between two different trends, this actually supports the idea of a qualitative difference between reliable and unreliable cells.

(I'm not sure this point is actually that important. The exponential fit seems to me to have no relevance to the final conclusions and the exponential relation hints at some kind of deeper mathematical truth which I suspect isn't real - I suggest just removing this, unless I've missed something.)

9. Analysis of data from Lin et al 2014 - Fig S7D - x-axis says "firing rates" but the data from Lin et al 2014 was calcium imaging - where do the firing rates come from?

10. In the model in Fig 3 the authors show how the models can recover the correct ratio of reliable:unreliable cells and other aspects from Fig. 2. It would be helpful to show (perhaps in a supplemental figure) if the models reproduce other aspects of the imaging data, e.g. the CDF of reliability, response amplitude, probability of overlap as in Fig. 2. Do they also follow a gamma distribution? Since these are curves rather than single metrics, it wouldn't be practical to show this for every model, but they could show some example curves (e.g., their favorite best-fitting model plus some that fit badly). Or if every model fits a gamma distribution, perhaps they could plot the gamma fit parameters as the y-axis of graphs akin to the graphs in Fig 3C-G. This would show both that the models are accurate, and that the reliability statistics don't depend on the number of cells (124 KCs in the data, X KCs in the model)

11. There is some confusing/missing information in the Methods:

- For the rate coding model in Fig. 3, the authors refer to their previous publications but don't provide the equations here or other details like the # of KCs. To save the reader having to track back into previous papers, they should provide the relevant equations and details here. The equations are important to understand the parameter values given in Table S1

- The Methods provide contradictory information about how many claws were assigned to each KC - a Poisson distribution (p. 37, first paragraph under "Figure 3") or a Binomial distribution (p. 38, last paragraph before "Figure S7"). Which is it? And what is the justification for using these distributions - can they be fit to real data? I could not find the analysis in Sterling and Laughlin 2015 explaining how parameters were fit to claw numbers of PN noise (indeed Caron et al 2013 and Bhandawat et al 2007 don't seem to be cited in Sterling and Laughlin 2015)

12. Some points the authors might wish to consider for the Discussion

- on the "saturate the cloud" vs "move the cloud" strategies, the authors should consider Baltruschat et al 2021 eLife which suggests that training leading to long-term memory actually can change KC representations. So there could be "cloud moving" in flies too - but only with repeated training - consistent with the results from Chapuis and Wilson

- The authors may wish to discuss their results in the context of the discrimination vs. generalisation trade-off (eg Barak et al 2013 which the authors have already cited) - It's probably good for similar odors to activate the same reliable cells because this allows the animal to generalise to similar odors (which could actually be the same odor as the trained odor but with some sensory noise added).

Minor points

Fig S5 B, E - how can there be probabilities > 1? Perhaps I misunderstood these graphs - can the authors clarify?

"Indeed, when examined over all trials (Fig. 1D), a total of 29% of the 124 KCs were unreliable, whereas 6% of the 124 KCs were reliable, which, as expected, is similar to the 5% fraction on an average trial."

Would this be clearer if they say "examined over all trials of each odor"?

Fig 2 - gamma distribution - is this including only the cells that respond at least once? (ie ignores the silent cells?)

p. 36 - "When we are considering odor-pair AB, the cell's overlap would be the reliability of the cell for odor A." - surely this is only true when the cell's reliability for odor B is 100%? Or is this a typo and it's supposed to say "the cell's *reliability* would be the reliability of the cell for odor A."?

p. 4 "cell 3 with a reliability of 1" suggest "reliability of 1 out of 4"? "1" creates momentary confusion, sounds like a perfect score of 1.0

Fig 7 - the arrows are a bit hard to distinguish thickness. Could they make the thick lines thicker?

Fig S5 - consider labelling the red and black curves directly on the figure

Fig 3H - the graded color bar on the right doesn't show up as graded on the MacOS Preview App (but it's OK on Adobe Acrobat) - might want to double check this before publication

Fig 3H legend "Darker colors indicate lower amounts of noise" - should this be "higher"

Fig 4B,C - should the y-axis labels say "KC similarity score" rather than "reliable KCs" and "unreliable KCs"?

Citations:

- Given the use of both mouse and fly, perhaps the authors would like to cite this recent review comparing the two systems? Endo and Kazama 2022 Curr Opin Neurobio

- one of the bioRxiv preprints cited has now been published (Abdelrahman et al 2021 PNAS)

Typos

p. 8 "Although, it does not capture a neuron's inner workings" - delete comma

p. 18 "other studies have shown at least three ways in which variability can improve an animal's fitness." - is this supposed to be "two ways"? the following text says (1) variability expands the search space, (2) stochastic resonance, with the authors' new idea of improving discrimination as the third way.

Missing equation number ?? on page 39, Figure 6.

p. 10 "In other words, as overlap increases (x-axis in Figs. 4E-F), the reliability (y- axis) increases proportionally: fly: r = 0.87, mouse: r = 0.91. Conversely, for dissimilar odor pairs (Figs. 4G-H), as overlap" - the panel references seem to be off by one? Same on p. 36 ("Figure 4E-H" - should be D-G)

---

## [Decision Letter · Decision Letter 2]

21 Apr 2023

Dear Dr Srinivasan,

Thank you for your patience while we considered your revised manuscript entitled "Stochastic coding: a conserved feature of odor representations and its implications for odor discrimination" for publication as a Research Article at PLOS Biology. This revised version of your manuscript has been evaluated by the PLOS Biology editors, the Academic Editor and the two original reviewers.

Based on the reviews, we are likely to accept this manuscript for publication provided you satisfactorily address the data and other policy-related requests stated below.

We expect to receive your revised manuscript within two weeks. 

*Published Peer Review History*

*Press*

Sincerely,

Ines

--

Ines Alvarez-Garcia, PhD

Senior Editor,

ialvarez-garcia@plos.org,

PLOS Biology

ETHICS STATEMENT:

-- Please state in the metadata of the manuscript that your manuscript does requires an ethics statement and please include in the statement the approval number.

Fig. 1B-I; Fig. 2C-H; Fig. 3B-G; Fig. 4C-H; Fig. 5A-G; Fig. 6A-F; Fig. 7A-F; Fig. S1A-D; Fig. S2A-D; Fig. S3A-H; Fig. S4A-G; Fig. S5A-F; Fig. S6A-F; Fig. S7A-J; Fig. S8A, B; Fig. S9A-F; Fig. S10A-E and Fig. S11A-E

Please also ensure that figure legends in your manuscript include information ON WHERE THE UNDERLYING DATA CAN BE FOUND, and ensure your supplemental data file/s has a legend.

Reviewers' comments

Rev. 1:

The authors have answered all the points raised in the previous review with reanalysis, rewriting and editing. The extensive responses to reviewers is clarifies things significantly, and I have no further concerns or objections.

Rev. 2:

The authors have done a thorough job of responding to my comments and I'm happy for the manuscript to be published. I just have two minor points about figure formatting.

---

## [Editor Report · Decision Letter 3]

21 Aug 2023

Dear Dr Srinivasan,

Thank you for the submission of your revised Research Article entitled "Effects of stochastic coding on olfactory discrimination in flies and mice" for publication in PLOS Biology. On behalf of my colleagues and the Academic Editor, Richard Benton, I am delighted to let you know that we can in principle accept your manuscript for publication, provided you address any remaining formatting and reporting issues. These will be detailed in an email you should receive within 2-3 business days from our colleagues in the journal operations team; no action is required from you until then. Please note that we will not be able to formally accept your manuscript and schedule it for publication until you have completed any requested changes.

PRESS

Sincerely, 

Ines

--

Ines Alvarez-Garcia, PhD

Senior Editor

PLOS Biology
